# On student-teacher deviations in distillation: does it pay to disobey?

## Abstract

Knowledge distillation has been widely-used to improve the performance of a "student" network by hoping to mimic soft probabilities of a "teacher" network. Yet, for self-distillation to work, the student *must* deviate from the teacher in some manner (Stanton et al., 2021). What is the nature of these deviations, and how do they relate to the generalization gains of distillation? To investigate these questions, we first conduct a variety of experiments across image and language classification datasets. One of our key observations is that in a majority of our settings, the student underfits points that the teacher finds hard. We also find that student-teacher deviations during the *initial* phase of training are *not* crucial to see the benefits of distillation — simply switching to distillation in the middle of training can recover much of its gains. We then provide two parallel theoretical perspectives of these deviations, one casting distillation as a regularizer in eigenspace, and another as a denoiser of gradients. In both views, we argue how student-teacher deviations emerge, and how they relate to generalization in the context of our experiments. Our analysis also bridges fundamental gaps between existing theory and practice by focusing on gradient descent and avoiding label noise assumptions.

## 1 Introduction

*Distillation* (Bucilă et al., 2006; Hinton et al., 2015) has emerged as a highly effective model compression technique, wherein one trains a small "student" model to match the predicted soft label distribution of a large "teacher" model, rather than match one-hot labels. An actively developing literature has sought to explore applications of this technique to various settings (Radosavovic et al., 2018; Furlanello et al., 2018; Xie et al., 2019), design more effective variants of the above recipe (Romero et al., 2015; Anil et al., 2018; Park et al., 2019; Beyer et al., 2022), and better understand theoretically when and why distillation is effective (Lopez-Paz et al., 2016; Phuong & Lampert, 2019; Mobahi et al., 2020; Allen-Zhu & Li, 2020; Menon et al., 2021; Dao et al., 2021).

On paper, distillation intends to transfer the teacher's soft probabilities over to the student. However, Stanton et al. (2021) challenge this premise: they show there is often a mismatch of student and teacher probabilities, and in fact, that a *greater* mismatch is correlated with *better* student performance. Indeed, in the *self-distillation* setting (Furlanello et al., 2018; Zhang et al., 2019) — where the student and teacher architectures are identical — some form of deviation (in the representation, if not in the probabilities) is *necessary* for the student's generalization to supercede the teacher.

In this work, we are interested in better characterizing these deviations in probabilities, and in understanding how they play a role in the student outperforming the teacher. In the first half of the paper, we conduct experiments characterizing *what* kind of deviations exist between the teacher and the student, and *which* deviations are relevant for better generalization. In the second half, we provide two complementary theoretical perspectives on *how* distillation can induce such deviations, and *why* that can subsequently aid generalization. More concretely, our *key* contributions are as follows:

(i) **What deviations exist?** Across various architectures (ResNet56, ResNet20, MobileNet, and RoBERTa), and image/language classification data (CIFAR100, TinyImageNet, CIFAR10, GLUE). we empirically demonstrate (§3.1) that the *the student tends to underfit on "hard" points for the teacher* (Fig 1a) in terms of the final probabilities learned by both models.

(ii) **Which deviations matter?** We find (§3.2) that it is possible to switch from one-hot loss in the middle of training to distillation loss and (a) still recover a considerable fraction of distillation's

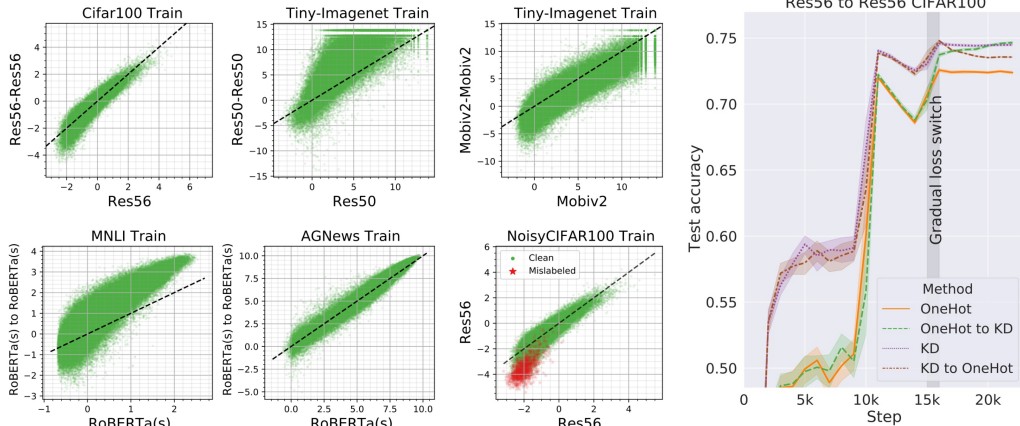

(a) Teacher-student logit plots for self-distillation.

(b) Effect of loss-switching.

Figure 1: **Left**: **Deviation in probabilities of (one-hot trained) teacher vs. (self-distilled) student:** For each training sample $(x, y)$, we plot $\phi(p^{\text{te}}_{y^{\text{te}}}(x))$ versus $\phi(p^{\text{st}}_{y^{\text{te}}}(x))$ for logit transformation $\phi(u) = \log[u/(1-u)]$ and teacher predicted label $y^{\text{te}}$. We consistently find that the distilled student predictions deviate from the $X = Y$ line (dashed) with teacher's "hard" points (small $X$) being *underfit* by the student ($Y \leq X$). This hints at distillation acting as a regularizer. **Right**: **Effect of late loss-switching:** In CIFAR-Resnet56 self-distillation, we switch the loss (gradually over the course of a few steps) late during training and find that switching to distillation (OneHot to KD line) recovers nearly all the gains of distillation (KD). This suggests that the initial phase of training is not critical for distillation to help.

gains (Fig 1b), (b) and also recover the final-epoch underfitting behavior on TinyImageNet and CIFAR100. Thus, we conclude that any student-teacher deviations unique to the early phase of training – such as those proposed in Allen-Zhu & Li (2020); Jha et al. (2020) – are *not* by themselves adequate to explaining the success of distillation, but the underfitting may be.

Next, we ask **how deviations arise** and **why they help**.

(iii) **Eigenspace view:** We provide a counterpart of the seminal result of Mobahi et al. (2020) — which demonstrates distillation as a regularizer in a *non-gradient-descent* setting — for the gradient descent setting for linear regression (Theorem 4.1). We propose this view as a way to understand the empirically observed underfitting in distillation. Besides providing a much simpler proof and a more practically relevant version of Mobahi et al. (2020), our result also formalizes existing empirical intuition about the importance of early-stopping in distillation (Dong et al., 2019; Cho & Hariharan, 2019; Ji & Zhu, 2020; Wang et al., 2022).

(iv) **Gradient space view.** As a complementary viewpoint, we formalize distillation as a *gradient denoiser* in the presence of class similarities (Theorem 4.2). We propose this view as a way to understand our empirical observations on loss-switching. Importantly, unlike prior work (Menon et al. (2021)), we show how denoising can occur even when the data is perfectly classifiable and has no inherent label noise.

(v) **A unified view.** We informally unify these two views, thus painting a more coherent picture of two disjoint lines of existing theories (Mobahi et al. (2020) vs. Menon et al. (2021)).

Overall, we hope that our discussion helps bridge the gap between existing theoretical understanding and empirics in distillation by (a) making more practical assumptions than existing theories, and (b) making connections to various empirical observations. Our findings also suggest that *not* matching the teacher probabilities exactly can be a good thing, which future empirical work on distillation may want to be mindful of.

## 2 BACKGROUND AND NOTATION

Our interest in this paper is *multiclass classification* problems. This involves learning a *classifier* $h\colon \mathcal{X} \to \mathcal{Y}$ which, for input $x \in \mathcal{X}$, predicts the most likely label $h(x) \in \mathcal{Y} = [K] \doteq \{1, 2, \ldots, K\}$. Such a classifier is typically implemented by computing *logits* $\mathbf{f}\colon \mathcal{X} \to \mathbb{R}^K$ that score the plausibility of each label, and then computing $h(x) = \operatorname{argmax}_{y \in \mathcal{Y}} f_y(x)$. In neural models, these logits are parameterised as $\mathbf{f}(x) = \mathbf{W}^\top \mathbf{Z}(x)$ for learned weights $\mathbf{W} \in \mathbb{R}^{D \times K}$ and embeddings $\mathbf{Z}(x) \in \mathbb{R}^D$. One may learn such logits by minimising the *empirical loss* on a training sample $S \doteq \{(x_n, y_n)\}_{n=1}^N$:

$$R_{\mathrm{emp}}(\mathbf{f}) \doteq \frac{1}{N} \sum_{n \in [N]} \mathbf{e}(y_n)^\top \ell(\mathbf{f}(x_n)), \tag{1}$$

where $\mathbf{e}(y) \in \{0,1\}^K$ denotes the *one-hot encoding* of $y$, $\ell(\cdot) \doteq [\ell(1, \cdot), \ldots, \ell(K, \cdot)] \in \mathbb{R}^K$ denotes the *loss vector* of the predicted logits, and each $\ell(y, \mathbf{f}(x))$ is the loss of predicting logits $\mathbf{f}(x) \in \mathbb{R}^K$ when the true label is $y \in [K]$. Typically, we set $\ell$ to be the softmax cross-entropy $\ell(y, \mathbf{f}(x)) = -\log p_y(x)$, where $\mathbf{p}(x) \propto \exp(\mathbf{f}(x))$ is the *softmax* transformation of the logits.

Equation 1 guides the learner via one-hot targets $\mathbf{e}(y_n)$ for each input. *Distillation* (Bucilă et al., 2006; Hinton et al., 2015) instead guides the learner via a target label distribution $\mathbf{p}^{\mathsf{te}}(x_n)$ provided by a *teacher*, which are the softmax probabilities from a distinct model trained on the *same* dataset. In this context, the learned model is referred to as a *student*, and the training objective is

$$R_{\mathrm{dist}}(\mathbf{f}) \doteq \frac{1}{N} \sum_{n \in [N]} \mathbf{p}^{\mathsf{te}}(x_n)^\top \ell(\mathbf{f}(x_n)). \tag{2}$$

One may also consider a weighted combination of $R_{\mathrm{emp}}$ and $R_{\mathrm{dist}}$ for algorithmic reasons, but we focus on the above objective in this paper since we are interested in understanding each objective individually. Compared to training on the one-hot labels, distillation often results in improved performance for the student (Hinton et al., 2015). Typically, the teacher model is of higher capacity than the student model; the performance gains of the student may thus informally be attributed to the teacher transferring rich information about the problem to the student. In such settings, distillation may be seen as a form of model compression. Intriguingly, however, even when the teacher and student are of the *same* capacity (a setting known as *self-distillation*), one may see gains from distillation (Furlanello et al., 2018; Zhang et al., 2019). The questions we explore in this paper are motivated by the self-distillation setting; however, for a well-rounded analysis, we will empirically study both the self- and cross-architecture-distillation settings.

## 3 A FINE-GRAINED LOOK AT TEACHER-STUDENT DEVIATIONS

Stanton et al. (2021) found that, contrary to the premise of distillation, more accurate students are poorer at matching the teacher probabilities. They quantified the student-teacher deviation by measuring the disagreement or the KL divergence between the student and teacher probabilities, in expectation over all points. Could the average deviation in probabilities be attributed merely to an *arbitrary* lack of precision in matching the probabilities during training? In the following section, through a closer study of *per-sample* relationship between the teacher and student predictions, we do *not* find this to be the case; rather, there are discernible patterns in student-teacher deviations.

### 3.1 WHAT DEVIATIONS EXIST: A PER-SAMPLE VIEW

Suppose we have teacher and distilled student models $\mathbf{f}^{\mathsf{te}}, \mathbf{f}^{\mathsf{st}}\colon \mathcal{X} \to \mathbb{R}^K$ respectively. We seek to visualize the deviations in the corresponding predicted probability vectors $\mathbf{p}^{\mathsf{te}}(x)$ and $\mathbf{p}^{\mathsf{st}}(x)$ for each $(x, y)$ in the train and test set, rather than in the aggregated sense as in Stanton et al. (2021). To visualize the deviations, we need a scalar summary of these vectors. An obvious candidate is the probabilities $(p_{y^\star}^{\mathsf{te}}(x), p_{y^\star}^{\mathsf{st}}(x))$ assigned to the ground truth label $y^\star$. However, since the student does not have access to the ground truth label, and is only trying to mimic the teacher, we examine deviations of probabilities of the *teacher's predicted label*, i.e., $(p_{y^{\mathsf{te}}}^{\mathsf{te}}(x), p_{y^{\mathsf{te}}}^{\mathsf{st}}(x))$ where $y^{\mathsf{te}} \doteq \operatorname{argmax}_{y' \in [K]} p_{y'}^{\mathsf{te}}(x)$. To make patterns easier to detect, we further perform a monotonic logit transformation $\phi(u) = \log [u/(1-u)]$ that produces real-values in $(-\infty, +\infty)$. Thus, we compare $\phi(p_{y^{\mathsf{te}}}^{\mathsf{te}}(x))$ and $\phi(p_{y^{\mathsf{te}}}^{\mathsf{st}}(x))$ for each train and test sample $(x, y)$.

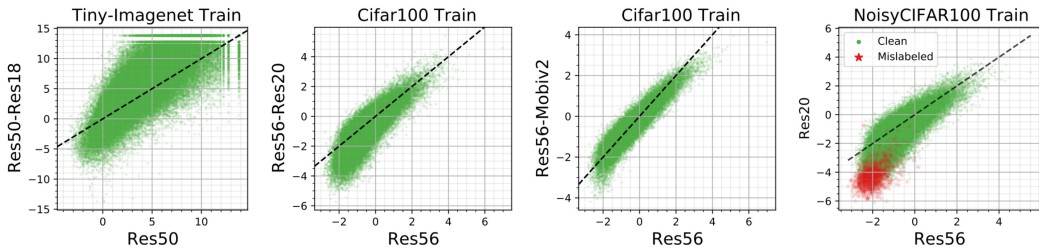

Figure 2: **Teacher-student logit-transformed probability plots under cross-architecture distillation settings:** In the last plot, we find that underfitting is clearly beneficial in deprioritizing noisy train labels, demonstrating that even a smaller student can go beyond a larger teacher.

We report a scatter plot of $\phi(p_{y^{te}}^{te}(x))$ ($X$-axis) vs. $\phi(p_{y^{te}}^{st}(x))$ ($Y$-axis) on the training set for both various self-distillation settings in Figures 1a and cross-architecture distillation settings in Fig 2. In all plots, the dashed line indicates the $X = Y$ line. All values are computed at the end of training. The tasks considered include image classification benchmarks, namely CIFAR10, CIFAR-100, Tiny-ImageNet, and text classification tasks from the GLUE benchmark (e.g., MNLI (Williams et al., 2018), AGNews (Zhang et al., 2015). See Appendix C.1 for details on the experimental hyperparameters. Many additional plots are presented in §C.2.1.

**Distilled students tend to underfit hard samples**. Our main finding is that in most of our cases, the student underfits a subset of points (i.e., $Y \leq X$ in the scatter plot) that are typically hard for the teacher (i.e., for small values of $X$). We note that this pattern is particularly strong on test data, and appears in the test data even in certain exceptions where it does not appear in the training data (see §C.2.1, e.g., CIFAR-10 plots Fig 7). We also note that in the language datasets, typically there is high deviation for harder points, involving both underfitting and overfitting, along with low deviation for easier points (i.e., large $X$), typically involving only overfitting (as can be seen in AGNews Fig 1 and in §C.2.1 Fig 8, Fig 9). These patterns however break down for cross-architecture distillation in language datasets. See §C.2.1 for a discussion of the exceptions.

**Why is underfitting good?** To better understand this, consider the setting where we perform distillation when a portion of CIFAR100 one-hot labels are mislabeled. In Fig 1a and Fig 2, we observe that the mislabeled points the student underfits the teacher's probabilities on *all* mislabeled points, implying that the student somehow *denoises* the data. *This forms the core of our intuition for why underfitting can be helpful*. As an aside, we note that even though the student denoises a portion of the data, this may not be reflected in the overall accuracy numbers when the student is smaller (Table 2). This brings out the subtlety that *smaller student models can go beyond a larger teacher in ways that are not apparent through mere accuracy on the whole dataset*. We provide a more formal view of the underfitting in §4.1 where we present distillation as a form of regularization. Having said that, we must note that underfitting may not always correspond to improved generalization (cf. §C.2.1, Table 2), just like regularization of any form may not always be helpful.

**Evolution of teacher-student deviations.** While we have looked at deviations between the *final* probabilities learned by the models, it is also instructive to look at how these deviations evolve as student training progresses. Concretely, in Fig 3a (and §C.2.3 Fig 13) we present a series of scatter plots for CIFAR100 (and Tiny-Imagenet) settings where the $X$ axis is always fixed to be the *final* (logit-transformed) teacher probabilities while the $Y$ axis corresponds to snapshots of the distillation-trained and the one-hot trained model at various instances. Here, we find significant differences between the scatter plots early on during training. In particular, the distilled model has converged much faster to its final values than the one-hot model. We also see from the left-most snapshot that the student has prioritized its easier points (i.e., points with large $X$ are close to the $X = Y$ line), while for the one-hot model, the easiest points are not as prioritized (i.e., points with large $X$ are farther from $X = Y$ compared to other points).

## 3.2 Which deviations matter: The effect of loss switching

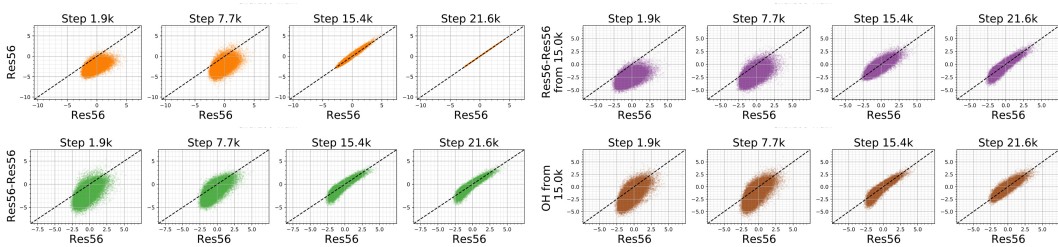

(a) One-hot and self-distillation.      (b) Loss-switching to distillation/one-hot at $15k$ steps.

Figure 3: **Evolution of teacher-student logit plots over various steps of training for CIFAR100 ResNet56 self-distillation setup:** On the **left**, we present plots for one-hot training (**top**) and distillation (**bottom**). There is a stark difference in the plots early on during training (at $1.9k$ steps): the student has converged to the $X = Y$ line much faster, fitting "easier" points more substantially. On the **right**, we present similar plots for experiments from Section 3.2, with the loss switched to distillation (**top**) and one-hot (**bottom**) at $15k$ steps. From the last two plots in each, we find that switching to distillation "unlearns" hard points, while switching to one-hot "fixes" the underfitting.

**Motivation.** Prior works suggest that student-teacher deviations in the very initial part of training are critical for why distillation helps: it conditions the network into a more favorable representation Allen-Zhu & Li (2020) or basin early on (Jha et al., 2020), or induces faster convergence (Phuong & Lampert, 2019) or perhaps, induces a favorable learning order (as is suggested by our own experiment in Fig 3a). However, in this section, we question whether initial phase deviations are critical at all.

**Experiment design.** To test the relevance of early-phase deviations, we conduct *late distillation* experiments where we train a network with one-hot loss for a fair part (say, half) of training, and then switch to distillation loss (gradually). If distillation relied crucially on early phase deviations, late distillation on a one-hot-pretrained network should not result in substantial improvements.

**Observations.** We make three key observations here. First, from Figures 4 (and Sec C.2.3 Fig 14), we observe that replacing the first $1/4$th of training with one-hot is able to recover *all* of distillation's gains in our CIFAR100 and TinyImagenet settings; replacing the first $1/2$ of training, recovers a signifcant fraction of the gains. This happens despite the fact that when we switch to distillation, we have a much smaller learning rate (due to our chosen schedule, see Sec C.1). This implies that *early phase deviations are not a necessary requirement for distillation to help*.

Conversely, we also switch from distillation to one-hot in Fig 1b and Fig 16. Here we find that switching to one-hot *undoes* the gains that distillation has achieved thus far, given a sufficiently long one-hot training. *This suggests an inherent destructive effect in one-hot training that we formalize in Sec 4.2.* We provide a more nuanced discussion of these experiments in Sec C.2.3 (and Fig 15).

Our final observation concerns the underfitting behavior in Section 3.1. In particular, even though early-phase one-hot training has already fit all examples, we find that subsequent distillation "unlearns" the harder examples to force the typical underfitting behavior (See Fig 3b, top). Conversely, switching to one-hot *fixes* the underfitting already induced by an initial distillation phase. We argue that *this reinforces the relevance of the underfitting of the final probabilities to the working of distillation*.

## 4   FORMALIZING STUDENT-TEACHER DEVIATIONS: TWO PERSPECTIVES

To understand the above empirical observations regarding underfitting and loss-switching, and their connections to the benefits of distillation, we provide two complementary perspectives of distillation: an *eigenspace* perspective and a *gradient-space* view. These perspectives also aim to provide key clarifications about existing intuition in literature, and paint a more coherent picture of disjoint threads of theoretical research.

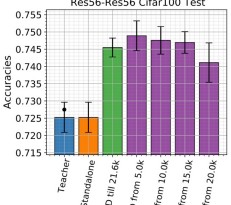 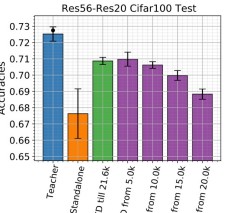 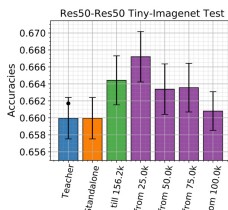 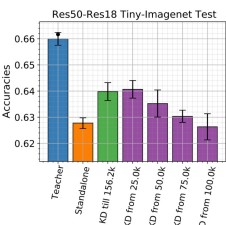

Figure 4: **Results of *loss-switching***: We report the performance of the teacher, one-hot student (Standalone; same as teacher in self-distillation), and distilled student (denoted by KD upto $T$ steps where $T$) and late-distilled student (denoted by KD from $T_{\text{switch}}$ step). We find that switching to distillation in the middle can recover large fractions of the gains of regular distillation.

### 4.1 THE EIGENSPACE VIEW: DISTILLATION AS A REGULARIZER

The underfitting of hard points possibly suggests that distillation induces a form of regularization that deprioritizes certain complex features. Indeed, Mobahi et al. (2020) have proven that self-distillation acts as a form of sparsifier that focuses on simpler basis functions while ignoring more complex ones. However, their result is agnostic to the biases of gradient descent.

We bridge the gap between this intuition and practice, by showing that a similar sparsification effect emerges under gradient descent training of linear models, thanks to the inherent bias of gradient descent in converging at varied rates along various eigendirections. Concretely, we analyze a continuous-flow gradient descent model on a linear regression setting with early-stopping, a typical design choice in distillation practice (Dong et al., 2019; Cho & Hariharan, 2019; Ji & Zhu, 2020). Consider an $n \times p$ dataset $\mathbf{X}$ (where $n$ is the number of samples, $p$ the number of parameters) with target labels $\mathbf{y}$. Assume that the Gram matrix $\mathbf{X}\mathbf{X}^\top$ is invertible. Note that this setting includes overparameterized scenarios ($p > n$) such as when $\mathbf{X}$ corresponds to the linearized (NTK) features of neural networks (Jacot et al., 2018; Lee et al., 2019). Then, a standard calculation reveals that the weights learned at time $t$ under gradient descent on the loss $(1/2) \cdot \|\mathbf{X} - \mathbf{y}\|^2$ can be written as:

$$\boldsymbol{\beta}(t) = \mathbf{X}^\top (\mathbf{X}\mathbf{X}^\top)^{-1} \mathbf{A}(t)\mathbf{y} \qquad \text{where } \mathbf{A}(t) := \mathbf{I} - e^{-t\mathbf{X}\mathbf{X}^\top}. \tag{3}$$

Intuitively, $\mathbf{A}$ is a "sparsifying" matrix that skews down the weight assigned to an eigendirection of eigenvalue $\lambda$ by the value $1 - e^{-\lambda t}$. As $t \to \infty$ this factor goes to 1 for all directions, thus becoming irrelevant; but for any finite $t > 0$, the topmost direction would have a larger factor than the rest, thus producing some sort of sparsification. Our argument is that, while standard gradient-descent already has an implicit sparsification effect, *distillation further amplifies the sparsification effect*. Consider a setting where the teacher is trained to time $T^{\text{te}}$. Through a simple calculation, the weights of the student model can be written as:

$$\tilde{\boldsymbol{\beta}}(\tilde{t}) = \mathbf{X}^\top (\mathbf{X}\mathbf{X}^\top)^{-1} \tilde{\mathbf{A}}(\tilde{t})\mathbf{y} \qquad \text{where } \tilde{\mathbf{A}}(\tilde{t}) := \mathbf{A}(t)\mathbf{A}(T^{\text{te}}). \tag{4}$$

Akin to Mobahi et al. (2020), one can argue that the sparsifier $\tilde{\mathbf{A}}$ corresponding to the student is more skewed towards the top eigenvectors than the teacher. More formally:

**Theorem 4.1.** *Let $\alpha_k(t), \tilde{\alpha}_k(t)$ be the eigenvalues of the $k$'th top eigendirection in $\mathbf{A}(t)$ and $\tilde{\mathbf{A}}(t)$ respectively. Let $\lambda_k$ be the corresponding eigenvalue of the Gram matrix $\mathbf{X}\mathbf{X}^\top$. For any two indices $k_1 < k_2$ such that $\lambda_{k_1} > \lambda_{k_2}$ (assuming such a pair exists), and for any $\alpha_{k_1}^\star \in (0,1)$ (a required "convergence lower bound" on the higher eigenvector), there exists a choice of distillation hyperparameters $T^{\text{te}}, \tilde{t}$ such that (a) $\tilde{\alpha}_{k_1}(\tilde{t}) \geq \alpha_{k_1}^\star$, and (b) for **any** choice of time $t$ for the standalone early-stopped model satisfying $\alpha_{k_1}(t) \geq \alpha_{k_1}^\star$, the components along the lower eigenvectors satisfy:*

$$\underbrace{\tilde{\alpha}_{k_2}(\tilde{t})}_{\substack{\text{Student's component} \\ \text{along lower eigenvector}}} \quad < \quad \underbrace{\alpha_{k_2}(t)}_{\substack{\text{Teacher's component} \\ \text{along lower eigenvector}}}. \tag{5}$$

The result says that the student relies less on the bottom eigendirections than the teacher if we compare them at any instant when they have both converged equally well along the top eigendirections (i.e.,

they have sufficiently fit the "nice" directions). *Crucially, this implies that the student traverses a different part of the parameter space, thus capturing a different bias than the teacher. In Sec C.4, we empirically verify that the student indeed amplifies the sparsification bias of the teacher, in a more general non-linear setting involving cross-entropy loss and a neural network.*

**Connection to underfitting.** The connection between how distillation amplifies regularization and the underfitting in §3.1 is best verified through our experiments on noisy CIFAR100 experiments from Fig 1a and Fig 2. First, it is well-known (Li et al., 2020; Dong et al., 2019; Arpit et al., 2017; Kalimeris et al., 2019) that the inherent bias of early-stopped gradient-descent fits noisy subset of the data more slowly than the clean data because noisy labels correspond to bottom eigendirections. This is indeed the case since mislabeled points come lowest in terms of teacher probabilities (small $X$ values). But our insights is that the student *exaggerates* this effect by assigning further lower probabilities to the mislabeled data ($Y < X$). This exaggeration must be the manifestation of how distillation amplifies the sparsification in the eigenspace.

## 4.2 THE GRADIENT-SPACE VIEW: DISTILLATION AS A GRADIENT DENOISER

While the previous perspective gives us an abstract theory of how distillation can induce regularization, it may not be obvious *when* this regularization can be helpful besides in settings with label noise. In fact, if ignoring the bottom eigenvectors led to ignoring non-noisy "tail" datapoints, this can even be hurtful (Feldman, 2019). Below, we formalize how distillation can help by extending a line of work analyzing distillation on noisy datasets with class similarities (Menon et al., 2021; Dao et al., 2021; Ren et al., 2022). But crucially, we show how distillation can help by *denoising* even in a perfectly classifiable dataset with class similarities, and *no explicit label noise*.

**A concrete example.** To make our discussion easier, consider a concrete $K$-class classification dataset, where the $i$th datapoint's features can be written as a $K$-"channel" input $\mathbf{x}_i = (\mathbf{x}_1^{(i)}, \ldots, \mathbf{x}_K^{(i)})$ where each channel $\mathbf{x}_k^{(i)} \in \mathbb{R}^D$ is $D$-dimensional. Assuming a uniform distribution over $K$ classes, given a label $y$, we generate $\mathbf{x}_y$ from a Gaussian $\mathcal{N}(0, \mathbf{I}/D)$ truncated to the support $\mathbf{x}_y \cdot \boldsymbol{\mu}_{y_i}^\star = \tau$ for a pre-defined $\tau > 0$ and a "ground truth" class vector $\boldsymbol{\mu}_k^\star$. We also assume that for point $\mathbf{x}_i$, we pick a non-target "similar" class $z_i$ for which we pick $\mathbf{x}_{z_i}$ from a Gaussian $\mathcal{N}(0, \mathbf{I}/D)$ truncated to the support $\mathbf{x}_{z_i} \cdot \boldsymbol{\mu}_{z_i}^\star = \tau/2$. All other co-ordinates are zero for $\mathbf{x}_i$. We consider a simplified linear architecture whose $K$-dimensional output is of the form $\mathbf{f}(\mathbf{x}) = (\mathbf{w}_1 \cdot \mathbf{x}_1, \ldots, \mathbf{w}_K \cdot \mathbf{x}_K)$ where the $k$'th node acts only on the $k$'th channel. Observe that this is a perfectly classifiable dataset with deterministic labels since if we set the model weights to be $\mathbf{w}_k = \boldsymbol{\mu}_k^\star$, we obtain that for the corresponding logits $\mathbf{f}^\star$, $\arg\max_k f_k^\star(\mathbf{x}) = y^\star$. In other words, the ground truth probabilities are one-hot, which can be recovered by computing softmax($\{\alpha \cdot \mathbf{f}_k(\mathbf{x})\}_{k=1}^K$) as $\alpha \to \infty$. Thus, it may seem that training on the one-hot loss is the wisest choice in this setting.

However, we show that distillation's gradients (a) are more optimal than one-hot gradients and also (b) more optimal than the teacher's weights themselves. To formalize this, let cos-sim$(\cdot, \cdot)$ denote the cosine-similarity between two vectors. Then:

**Theorem 4.2.** *(informal; see Thm B.4) Consider a student whose weights satisfy $\mathbf{w}_k = \alpha \boldsymbol{\mu}_k^\star$. Consider an imperfect teacher with weights such that $\|\mathbf{w}_k^{te}\| = \alpha_k^{te}$ and cos-sim$(\mathbf{w}_k^{te}, \boldsymbol{\mu}_k^\star) \geq 1 - \epsilon$. Let $\mathbf{v}_k$ be the one-hot update on node $k$ of the student, and $\tilde{\mathbf{v}}_k$ the distillation update, both under cross-entropy loss. Then, for $\alpha \ll \alpha^{te}$ and $\alpha^{te} \leq O(\frac{1}{\tau} \log K)$, and sufficiently large $\tau, D$:*

$$\underbrace{cos\text{-}sim(\tilde{\mathbf{v}}_k, \boldsymbol{\mu}_k^\star)}_{\textit{Quality of distillation update}} > \max \left( \underbrace{cos\text{-}sim(\mathbf{v}_k, \boldsymbol{\mu}_k^\star)}_{\textit{Quality of one-hot update}}, \underbrace{cos\text{-}sim(\mathbf{w}_k^{te}, \boldsymbol{\mu}_k^\star)}_{\textit{Quality of teacher weights}} \right) \tag{6}$$

There are three salient aspects to this result. First, this demonstrates that one-hot gradients can be inherently sub-optimal at any timestep in training *even if the model is initialized at an optimal solution*. Conversely, distillation can help improve these gradients *even if the teacher is imperfect*. Finally, distillation gradients can themselves be "more perfect" than the imperfect teacher, thus formalizing how the student can go beyond the teacher.

To understand the intuition behind this result, imagine that at some point of training, we have recovered a good $\mathbf{f}$ such that $\mathbf{f} \approx \alpha \cdot \mathbf{f}^\star$ (for some finite scaling factor $\alpha$). Now as we train longer, we ideally hope that (a) $\alpha$ continues increasing towards $\infty$ to recover the one-hot ground truth probabilities, and (b) the approximation of the ground truth logits continues to hold throughout.

This is however not the case with the one-hot updates. Consider any point $(\mathbf{x}, y)$ where $\mathbf{f}_k^\star(\mathbf{x})$ is high for some $k \neq y$ i.e., the point has similarities to a non-target class. On such a point, the ideal dynamics of increasing $\alpha$ requires that we *increase the logit on the non-target node* $f_k(\mathbf{x})$. But the one-hot update would attempt to suppress this logit, since the target probability for this node is $0$. We argue that this is a *destructive gradient* since it directs the model in the opposite direction of the ideal dynamics. Notably, this manifests despite the lack of inherent label noise in the training set.

Distillation on the other hand, can flip the sign of these gradients (assuming the teacher is early-stopped), thus denoising the gradients — even if there is no explicit noise in the data. Notably, this sign-flipping provides a "fresh example" for the non-target node $k$, which is a privilege unavailable to the teacher. This is crucial to prove that the student is more denoised than the the teacher.

**Connection to loss-switching.** The gradient-space view gives us a way to make sense of the findings in §3.2. In particular, this tells us why each gradient in itself may be constructive (in the case of distillation) or destructive (in the case of one-hot). Thus, switching to distillation (or one-hot) even in the middle of training can still be helpful (or hurtful), given sufficient time after the switch.

### 4.3 Unifying the two perspectives

We now discuss how various themes in existing distillation theory can be understood within both of the above perspectives. Through this, we hope to piece together a more coherent and unified picture of the two lines of work we build upon: Mobahi et al. (2020) and Menon et al. (2021).

**Dark knowledge and denoising.** Distillation is said to benefit by implicitly introducing "dark knowledge" in the form of new information absent in the observed labels of the original datasets (Hinton et al., 2015). In both views, we can view dark knowledge as a form of recovering denoised features:

(i) In the gradient-space view, the dark knowledge comes from the non-target logits, which allows us to denoise the gradients during distillation.

(ii) In the eigenspace view, we posit that the new information corresponds to knowledge of the top eigenvector(s) of the data. Crucially, *these eigenvectors are independent of labels* and thereby independent of any label noise; hence these eigenvectors correspond to "denoised" features in the data, which distillation more readily relies on.

**On the gap between early-stopping and distillation.** In both our views, we see a subtle distinction between early-stopping and distillation:

(i) In the eigenspace perspective, although both induce a similar sparsification effect, distillation *amplifies* this effect.

(ii) In the gradient-space view, early-stopping would reduce the overall magnitude of destructive gradients accumulated by the teacher. But distillation amplifies this effect further by altogether flipping the sign of these gradients.

**Lack of fidelity in probabilities.** We can understand the underfitting phenomenon from both views:

(i) Recall that in the eigenspace view, the student's underfitting of teacher's probabilities can be seen as a consequence of ignoring the lower eigenvectors.

(ii) In the gradient-space view, the harder points of the teacher would have high similarity to other classes. Subsequently, under the one-hot loss, these points would experience heavy destructive gradients on the non-target logits. Since distillation would no longer suppress the non-target probabilities, those probabilities would increase, while the target probability would be underfit.

## 5 Relation to Existing Work

**Distillation as a probability matching process.** Distillation has been touted to be a process that benefits from matching the teacher's probabilities (Hinton et al., 2015). Indeed, many distillation algorithms have been designed in a way to more aggressively match the student and teacher functions (Czarnecki et al., 2017; Beyer et al., 2022). Theoretical analyses too rely on explaining the benefits of distillation based on a student that obediently matches the teacher's probabilities (Menon et al., 2021). However, as Stanton et al. (2021) demonstrate, an exact matching of probabilities can

be *anti-correlated* with performance improvements. Our work builds on this nuance as to why we may *desire* that the student deviate from the teacher.

**Understanding distillation via class similarities.** A long-standing intuitive explanation for why distillation helps is that the teacher's probabilities contain "dark knowledge" about class similarities (Hinton et al., 2015; Müller et al., 2019), which is the idea we formalize in §4.2. Several works (Menon et al., 2021; Dao et al., 2021; Ren et al., 2022; Zhou et al., 2021) have formalized this by assuming that the ground truth class memberships are inherently soft, and so the observed one-hot labels are "noisy". They argue that distillation reduces the student model's variance (in an abstract non-gradient-descent setting), assuming the student exactly matches the teacher. We demonstrate a novel form of denoising even in the absence of noisy labels (via a completely different technical argument). We also show how the student can outperform the teacher under gradient descent.

**Alternative theories of distillation.** Some works (Furlanello et al., 2018; Yuan et al., 2020; Tang et al., 2020) have argued that the "class similarity" hypothesis cannot be the sole explanation, because distillation can help even if the student is only taught information about the target probabilities (e.g., by smoothing out all non-target probabilities). This has resulted in various alternative hypothesis such as *faster convergence* (Phuong & Lampert, 2019; Rahbar et al., 2020; Ji & Zhu, 2020) (which we question in §3.2), *feature learning* (Allen-Zhu & Li, 2020) (which would not apply in our linear settings), and regularization, either in the sense of Mobahi et al. (2020) or in the sense of *instance-specific label smoothing* (Zhang & Sabuncu, 2020; Yuan et al., 2020; Tang et al., 2020). Our eigenspace perspective in §4.1 falls under this umbrella. Finally, we also refer the reader to Lopez-Paz et al. (2016); Kaplun et al. (2022) who theoretically study distillation in orthogonal settings.

**Early-stopping and knowledge distillation.** Early-stopping has received much attention in the context of distillation (Liu et al., 2020; Ren et al., 2022; Dong et al., 2019; Cho & Hariharan, 2019; Ji & Zhu, 2020). We closely build on Dong et al. (2019), who argue how early-stopping a gradient-descent-trained teacher can automatically denoise the labels due to regularization in the eigenspace. However, neither Dong et al. (2019) nor any of the other works provide a formal argument for why distillation can outperform the teacher. Note that Mobahi et al. (2020) claim early-stopping has a *densification* rather than a sparsification effect; but this holds only in their non-gradient-descent setting where the function is picked from a Hilbert space with $\ell_2$ regularization.

**Empirical studies of distillation.** Our study crucially builds on observations from (Stanton et al., 2021; Lukasik et al., 2021) demonstrating student-teacher deviations in an aggregated sense than a sample-wise sense sense. Other studies (Abnar et al., 2020; Ojha et al., 2022) investigate in what ways the student is *similar* to the teacher, beyond in terms of class probabilities e.g., out-of-distribution behavior, calibration, and so on. Deng & Zhang (2021) show how a smaller student can outperform the teacher if it was allowed better match the teacher on more data, which is orthogonal to our setting.

## 6 CONCLUSION

We empirically find that a distilled student tends to underfit points that the teacher finds hard. We also find that student-teacher deviations that occur in the initial phase of training are not critical for the success of distillation. We then formalized how deviations arise and can help the student. Specifically, in the context of gradient descent, distillation can act as a regularizer in eigenspace and can denoise gradients even in the absence of explicit label noise. Overall, our work provides a concrete argument for why certain deviations between the student and teacher are good, contrary to conventional understanding of distillation. We also bridge key gaps between theory and practice, and unify disjoint lines of work following Mobahi et al. (2020) and Menon et al. (2021).

There are several directions for future work. One question is whether we can study more practical distillation setups, such as in *semi-supervised* setups (Cotter et al., 2021). It would also be insightful to extend our eigenspace view in to multi-layered models where the eigenspace regularization effect may "compound" across layers. Another interesting direction is whether we can build on the gradient space view to design teacher-free training techniques that suppress "destructive" non-target gradients in one-hot training. Arguably, the most important implication is on current attempts to bridge the gap between student and teacher performance. Perhaps, these attempts would benefit from encouraging *some carefully-designed* gaps in the teacher and student probabilities.

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

## A    LIMITATIONS

We highlight a few key limitations to our results that may be relevant for future work to look at:

1. While we show underfitting across many datasets, we do not establish when this form of underfitting can help in situations besides label noise. Indeed, we find that, in our language datasets, underfitting can occur even in the absence of any improvement in generalization. Future work may explore more sophisticated measures of student-teacher deviation that correlates with the gains of distillation. We highlight further caveats of our empirical results in Sec C.2.1.

2. Both our formalizations are based on linear models. On the one hand, this implies that our results are general and fundamental in that they are not unique to neural network training. On the other, this means our result may not capture other interesting effects in the context of distillation (such as that of feature-learing as in Allen-Zhu & Li (2020)). Indeed, we hypothesize that the eigenspace regularization effect may compound across multiple layers of a neural network. This is an open direction for future work.

3. While our gradient-space view suggests why switching to one-hot can hurt, it does not explain why switching to one-hot for a very short amount of time increases the performance — even beyond distillation — in the case of ResNet50 self-distillation (cf. Sec C.2.3 Fig 16 and Fig 15). Analyzing this nuanced effect of distillation-to-one-hot switching is an interesting question for future work.

## B    EIGENSPACE VIEW

Below, we provide the proof for the sparsity levels of an early-stopped teacher and early-stopped student.

*Proof.* (of Theorem 4.1) For the early-stopped model, we need $1 - e^{-\lambda_{k_1} t} \geq \alpha_{k_1}^\star$. For any value of $t$ that satisfies this, we can lower bound $\alpha_{k_2}(t)$ as:

$$\alpha_{k_2}(t) = 1 - e^{-\lambda_{k_2} t} \tag{7}$$

$$= 1 - \left(e^{-\lambda_{k_1} t}\right)^{\frac{\lambda_{k_2}}{\lambda_{k_1}}} \tag{8}$$

$$\geq 1 - \left(1 - \alpha_{k_1}^\star\right)^{\frac{\lambda_{k_2}}{\lambda_{k_1}}} . \tag{9}$$

Now, recall that we need to show that *there exists* a set of distillation hyperparameters $\tilde{t}, T^{\text{te}}$ that satisfy our required bounds. We will show that such a set of hyperparameters exist under the simplification that $T^{\text{te}} = \tilde{t}$. Thus, for the student model, we need $(1 - e^{-\lambda_{k_1} \tilde{t}})(1 - e^{-\lambda_{k_1} \tilde{t}}) \geq \alpha_{k_1}^\star$. To achieve the largest skew, we should choose the smallest $\tilde{t}$, which is achieved when $(1 - e^{-\lambda_{k_1} \tilde{t}})^2 = \alpha_{k_1}^\star$. Plugging this into the expression for $\tilde{\alpha}_{k_2}(\tilde{t})$, we get:

$$\tilde{\alpha}_{k_2}(\tilde{t}) = (1 - e^{-\lambda_{k_2} \tilde{t}})^2 \tag{10}$$

$$= \left(1 - \left(e^{-\lambda_{k_1} \tilde{t}}\right)^{\frac{\lambda_{k_2}}{\lambda_{k_1}}}\right)^2 \tag{11}$$

$$= \left(1 - (1 - \sqrt{\alpha_{k_1}^\star})^{\frac{\lambda_{k_2}}{\lambda_{k_1}}}\right)^2 . \tag{12}$$

For simplicity, let us define the fraction $\kappa := \frac{\lambda_{k_2}}{\lambda_{k_1}}$. Putting the above two sets of equation together, we have:

$$\alpha_{k_2}(t) - \tilde{\alpha}_{k_2}(\tilde{t}) \leq 1 - \left(1 - \alpha_{k_1}^\star\right)^f - \left(1 - (1 - \sqrt{\alpha_{k_1}^\star})^f\right)^2 \tag{13}$$

$$\leq 1 - \left(1 - \alpha_{k_1}^\star\right)^f - 1 - ((1 - \sqrt{\alpha_{k_1}^\star})^2)^f + 2(1 - \sqrt{\alpha_{k_1}^\star})^f \tag{14}$$

$$\leq 2(1 - \sqrt{\alpha_{k_1}^\star})^f - ((1 - \sqrt{\alpha_{k_1}^\star})^2)^f - \left(1 - \alpha_{k_1}^\star\right)^f \tag{15}$$

$$\leq 2 - (1 - \sqrt{\alpha_{k_1}^\star}) \left((1 - \sqrt{\alpha_{k_1}^\star}))^f - \left(1 - \sqrt{\alpha_{k_1}^\star}\right)^f\right) \qquad \leq 0 \tag{16}$$

The last step follows from the power means inequality: we have that since $f < 1$, $\left(\frac{(1-\sqrt{\alpha_{k_1}^\star}))^f + (1+\sqrt{\alpha_{k_1}^\star}))^f}{2}\right)^f$ is at most $\left(\frac{(1-\sqrt{\alpha_{k_1}^\star})) + (1+\sqrt{\alpha_{k_1}^\star}))}{2}\right) = 1$. $\qquad\square$

## B.1 GRADIENT-SPACE VIEW

In this section, we prove Theorem 4.2 where we analyzed the quality of one-hot and distillation gradients in a setting with class similarity and zero label noise. We first recall the setting below.

**Setting.** We consider a $K$-class classification dataset, where the $i$th datapoint's features can be written as a $K$-"channel" input $\mathbf{x}_i = (\mathbf{x}_1^{(i)}, \ldots, \mathbf{x}_K^{(i)})$ where each channel $\mathbf{x}_k^{(i)} \in \mathbb{R}^D$ is $D$-dimensional. Assuming a uniform distribution over $K$ classes, given a label $y$, we generate $\mathbf{x}_y$ from a Gaussian $\mathcal{N}(0, \mathbf{I}/D)$ truncated to the support $\mathbf{x}_y \cdot \boldsymbol{\mu}_{y_i}^\star = \tau$ for $\tau > 0$ and for a pre-defined class vector $\boldsymbol{\mu}_k^\star$. We also assume that for point $\mathbf{x}_i$, we pick a non-target "similar" class $z_i$ for which we pick $\mathbf{x}_{z_i}$ from a Gaussian $\mathcal{N}(0, \mathbf{I}/D)$ truncated to the support $\mathbf{x}_{z_i} \cdot \boldsymbol{\mu}_{z_i}^\star = \tau/2$. All other co-ordinates are set to zero. We consider a simplified linear architecture whose $K$-dimensional output is of the form $\mathbf{f}(\mathbf{x}) = (\mathbf{w}_1 \cdot \mathbf{x}_1, \ldots, \mathbf{w}_K \cdot \mathbf{x}_K)$ where the $k$'th node acts only on the $k$'th channel.

**Notations**: Throughout the rest of the discussion, we fix some node $k$, and define $S_+ = \{(\mathbf{x}, y) \in S | y = k\}$ and $S_- = \{(\mathbf{x}, y) \in S | y \neq k, z = k\}$. Intuitively, $S_+$ is the subset of samples that provide "constructive" gradients for node $k$ under the one-hot loss, while $S_-$ is the subset that provides "destructive" gradients under the one-hot loss (i.e., gradients in the opposite direction of optimality). With an abuse of notation, we will write $\mathbf{x} \in S$ to denote an $\mathbf{x}$ such that $(\mathbf{x}, y) \in S$.

We will use $\mathbf{p}^{\text{te}}$ to denote a perfect teacher's probabilities, and $\mathbf{q}^{\text{te}}$ to denote an imperfect teacher's probabilities. For any channel $k$, and for any vector $\mathbf{w} \in \mathbb{R}^D$ of weights/gradients corresponding to that channel, we will use $\mathbf{w}^{\|}$ to denote the projection $\mathbf{w}^{\|} = (\mathbf{w} \cdot \boldsymbol{\mu}_k^\star)\boldsymbol{\mu}_k$, and $\mathbf{w}^\perp$ to denote the orthogonal vector, $\mathbf{w}^{\|} = \mathbf{w} - (\mathbf{w} \cdot \boldsymbol{\mu}_k^\star)\boldsymbol{\mu}_k$.

Also recall that $\alpha^{\text{te}}$ denotes the "scale" of the teacher's logits — think of this as a proxy for how long the teacher has been run to minimize one-hot loss. Similarly, $\alpha$ denotes the scale of the student model's logits. We now state some simplifying assumptions.

**Assumption B.1.** *We make the following simplifying assumptions:*

1. *We assume that we have a training dataset $S$ such that $S = mK$. The dataset is balanced in that for every $k$, $|S_+| = |S_-| = m$.*

2. *We assume $\|\boldsymbol{\mu}_k^\star\| = 1$.*

3. *We assume $\epsilon^{\text{te}} < 1$ and $\tau > 1$.*

4. *We assume $4\alpha^{\text{te}}\tau\epsilon^{\text{te}} \leq 1$.*

In Lemma B.1, we show that as long as both the teacher and the student are not trained for too long (and thus their scales $\alpha^{\text{te}}$ and $\alpha$ are low), the (perfect) teacher's supervision is "constructive" on node $k$ for all points in $S_-$ and $S_+$. This is crucial for us to show that distillation is able to denoise the one-hot gradients (which are destructive on $S_-$). In the subsequent Lemma B.2, we bound the difference in supervision of a perfect and imperfect teacher. Using this, we can show that despite imperfection, the supervision of the teacher can be constructive for node $k$ on both $S_-$ and $S_+$.

**Lemma B.1.** *Assuming $\alpha^{te}\tau \leq \log(K-2)$ and $\alpha^{te} - \alpha \geq \frac{2}{\tau}\log\frac{3}{1-\kappa}$, then for any $\mathbf{x} \in S_+ \cup S_-$,*
$\frac{p_k^{te}(\mathbf{x}) - p_k(\mathbf{x})}{p_k^{te}(\mathbf{x})} \geq \kappa.$

*Proof.* If the point belongs to $S_-$, then:

$$p_k(\mathbf{x}) = \frac{e^{\alpha\tau/2}}{e^{\alpha\tau} + e^{\alpha\tau/2} + K - 2} \quad \text{and} \quad p_k^{te}(\mathbf{x}) = \frac{e^{\alpha^{te}\tau/2}}{e^{\alpha^{te}\tau} + e^{\alpha^{te}\tau/2} + K - 2}. \tag{17}$$

Then, the ratio is :

$$\frac{p_k(\mathbf{x})}{p_k^{te}(\mathbf{x})} = \frac{e^{\alpha^{te}\tau/2} + 1 + (K-2)e^{-\alpha^{te}\tau/2}}{e^{\alpha\tau/2} + 1 + (K-2)e^{-\alpha\tau/2}}. \tag{18}$$

Since $\alpha^{te}\tau \leq \log(K-2)$, we can upper-bound the first two terms in the numerator by the last term. Hence,

$$\frac{p_k(\mathbf{x})}{p_k^{te}(\mathbf{x})} \leq \frac{3(K-2)e^{-\alpha^{te}\tau/2}}{(K-2)e^{-\alpha\tau/2}} \leq 3e^{(\alpha-\alpha^{te})\frac{\tau}{2}} \leq 1 - \kappa. \tag{19}$$

If the point belongs to $S_+$, then:

$$p_k(\mathbf{x}) = \frac{e^{\alpha\tau}}{e^{\alpha\tau} + e^{\alpha\tau/2} + K - 2} \quad \text{and} \quad p_k^{te}(\mathbf{x}) = \frac{e^{\alpha^{te}\tau}}{e^{\alpha^{te}\tau} + e^{\alpha^{te}\tau/2} + K - 2}. \tag{20}$$

The ratio is

$$\frac{p_k(\mathbf{x})}{p_k^{te}(\mathbf{x})} = \frac{1 + e^{-\alpha^{te}\tau/2} + (K-2)e^{-\alpha^{te}\tau}}{1 + e^{-\alpha\tau/2} + (K-2)e^{-\alpha\tau}}$$

Again, we can upper-bound each term in the denominator by the last term.

$$\frac{p_k(\mathbf{x})}{p_k^{te}(\mathbf{x})} \geq \frac{3(K-2)e^{-\alpha^{te}\tau}}{(K-2)e^{-\alpha\tau}} \leq 1 - \kappa.$$

$\square$

**Lemma B.2.** *Consider a perfect teacher model with weights $\alpha^{te}\boldsymbol{\mu}_k^\star$ and corresponding probabilities $\mathbf{p}^{te}(\cdot)$. Consider an $\epsilon^{te}$-imperfect teacher model with weights $\mathbf{w}_k$ such that $\|\mathbf{w}_k\| = \alpha^{te}$ and $\frac{\|\mathbf{w}_k^\perp\|^2}{\|\mathbf{w}_k^\|\|^2} \leq \epsilon^{te}$. Let the corresponding probabilities be given by $\mathbf{q}^{te}(\cdot)$.*

*Assuming $4\alpha^{te}\tau\epsilon^{te} \leq 1$, on any $\mathbf{x} \in S_+ \cup S_-$,*

$$|p_k^{te}(\mathbf{x}) - q_k^{te}(\mathbf{x})| \leq 12\alpha^{te}\tau \cdot \sqrt{\epsilon^{te}} \cdot p_k^{te}(\mathbf{x}). \tag{21}$$

*Proof.* We have $\|\mathbf{w}^\perp\|^2 \leq (\alpha^{te})^2 \frac{\epsilon^{te}}{1+\epsilon^{te}}$ and $\|\mathbf{w}^\|\|^2 \geq (\alpha^{te})^2 \frac{1}{1+\epsilon^{te}}$.

We have

$$\mathbf{w}_k \cdot \mathbf{x}_k = \|\mathbf{w}^\|\| \cdot (\mathbf{x}_k \cdot \boldsymbol{\mu}_k^\star) + \mathbf{w}^\perp \cdot \mathbf{x}_k^\perp. \tag{22}$$

We have $\|\mathbf{w}^\|\| \leq \alpha^{te}$ and $\|\mathbf{w}^\|\| \geq \alpha^{te}\frac{1}{\sqrt{1+\epsilon^{te}}} \geq \alpha^{te}(1 - \epsilon^{te})$. We also have $|\mathbf{x}_k \cdot \boldsymbol{\mu}_k^\star| \leq \tau$. Next, $\|\mathbf{w}^\perp\| \leq \alpha^{te}\sqrt{\frac{\epsilon^{te}}{1+\epsilon^{te}}} \leq \alpha^{te}\sqrt{\epsilon^{te}}$. Finally, $\lim_{D\to\infty} \|\mathbf{x}_k^\perp\| \leq 1$. Putting these together, we get:

$$|\mathbf{w}_k \cdot \mathbf{x}_k - \alpha^{te}(\mathbf{x}_k \cdot \boldsymbol{\mu}_k^\star)| \leq \alpha^{te}\tau\epsilon^{te} + \alpha^{te}\sqrt{\epsilon^{te}} \leq 2\alpha^{te}\tau\sqrt{\epsilon^{te}}. \tag{23}$$

Here, we've assumed $\epsilon^{te} < 1$ and $\tau > 1$.

The above gives us a bound on the differences in the logits. But to bound the probabilities themselves, note that due to the structure of the softmax, both the numerator and the denominator can either grow or diminish by a factor of $e^{2\alpha^{\text{te}}\tau\sqrt{\epsilon^{\text{te}}}}$. Thus, $e^{4\alpha^{\text{te}}\tau\sqrt{\epsilon^{\text{te}}}} \geq \frac{q_k^{\text{te}}(\mathbf{x})}{p_k^{\text{te}}(\mathbf{x})} \geq e^{-4\alpha^{\text{te}}\tau\sqrt{\epsilon^{\text{te}}}}$. We have $e^{-4\alpha^{\text{te}}\tau\sqrt{\epsilon^{\text{te}}}} \geq 1 - 4\alpha^{\text{te}}\tau\sqrt{\epsilon^{\text{te}}}$.

Also, given that $4\alpha^{\text{te}}\tau\epsilon^{\text{te}} \leq 1$, $e^{4\alpha^{\text{te}}\tau\epsilon^{\text{te}}} \leq 1 + e \cdot 4\alpha^{\text{te}}\tau\epsilon^{\text{te}}$. Rearranging this gives us our final inequality.

$\square$

First, we prove a simpler statement assuming that the teacher is perfect. We will build on this proof for an imperfect teacher.

**Lemma B.3.** *Consider a student model with weights $\mathbf{w}_k = \alpha\boldsymbol{\mu}_k^\star$ such that $\alpha \leq \frac{1}{\tau}\log(K-2)$. Let $\mathbf{u}_k$ denote the one-hot gradients on this model under the cross-entropy loss on the dataset $S$. Consider an perfect teacher model with weights $\mathbf{w}_k$ such that $\mathbf{w}_k = \alpha^{\text{te}}\boldsymbol{\mu}_k^\star$. Assume the teacher model is early-stopped in that $\alpha^{\text{te}} \leq \frac{1}{\tau}\log(K-2)$.*

*Assuming $\alpha^{\text{te}} \geq \frac{1}{\tau}\log 3$, let $\kappa > 0$ denote the level of early-stopping in the student, such that $\alpha^{\text{te}} - \alpha \geq \frac{2}{\tau}\log\frac{3}{1-\kappa}$. Let $\tilde{\mathbf{u}}_k$ denote the gradient under distillation loss with this teacher on the above student model.*

*Assume that the simplifying assumptions of Assumption B.1 hold. Then,*

$$\underbrace{\lim_{D\to\infty}\frac{\|\mathbf{u}_k^\|\|^2}{\|\mathbf{u}_k^\perp\|^2} \leq m\tau^2\left(1 - \frac{e^{-\alpha\tau}}{9}\right)}_{\textit{Upper-bound on quality of one-hot gradients}} \leq \underbrace{m\tau^2\left(1 + \kappa^2\right) \leq \lim_{D\to\infty}\frac{\|\tilde{\mathbf{u}}_k^\|\|^2}{\|\tilde{\mathbf{u}}_k^\perp\|^2}}_{\textit{Lower-bound on quality of distillation gradients}}. \tag{24}$$

*Proof.* We have assumed $|S_+| = |S_-| = m$. Note that for all $\mathbf{x}$ in $S_+$, $p_k(\mathbf{x}) = p_+$ for some constant $p_+ > 0$. Similarly, for all $\mathbf{x}$ in $S_-$, $p_k(\mathbf{x}) = p_-$ for some constant $p_+ > 0$.

Recall that the gradient descent update on node $k$ be written as $\mathbf{u}_k = \mathbf{u}_k^\| + \mathbf{u}_k^\perp$, where $\mathbf{u}_k^\perp \cdot \boldsymbol{\mu}_k^\star = 0$.

The one-hot gradient at node $k$ can be written as:

$$\mathbf{u}_k = \sum_{\mathbf{x}\in S_+}(1 - p_k(\mathbf{x}))\mathbf{x}_k - \underbrace{\sum_{\mathbf{x}\in S_-}p_k(\mathbf{x})\mathbf{x}_k}_{\text{destructive gradients}} \tag{25}$$

$$= (1 - p_+)\sum_{\mathbf{x}\in S_+}\mathbf{x}_k - p_-\left(\sum_{\mathbf{x}\in S_-}\mathbf{x}_k\right). \tag{26}$$

We project the sum total of all these gradients along $\boldsymbol{\mu}_k^\star$ to get

$$\|\mathbf{u}_k^\|\| = \mathbf{u}_k \cdot \boldsymbol{\mu}_k^\star = m\tau\left((1 - p_+) - p_-\right).$$

Next, we want to compute the total magnitude of all the gradients orthogonal to $\boldsymbol{\mu}_k^\star$, namely $\|\mathbf{u}_k^\perp\|$. Note that the projection of any sample $\mathbf{x}$ orthogonal to $\boldsymbol{\mu}_k^\star$ is essentially a $D-1$ multivariate zero-mean Gaussian with variance $1/D$ along each direction. Thus, the projection of $\mathbf{u}_k^\perp$ along each of these $D-1$ directions is a zero-mean Gaussian with variance equal to $\frac{m}{D}\left((1-p_+)^2 + p_-^2\right)$. Thus $\|\mathbf{u}_k^\perp\|^2$ is the sum of squared of $D-1$ of these random variables. Therefore,

$$\lim_{D\to\infty}\|\mathbf{u}_k^\perp\| = m\left((1-p_+)^2 + p_-^2\right).$$

Thus, the ratio can be written as:

$$\lim_{D \to \infty} \frac{\|\mathbf{u}_k^{\|}\|^2}{\|\mathbf{u}_k^{\perp}\|^2} = \frac{m^2 \tau^2 \left((1-p_+) - p_-\right)^2}{m \left((1-p_+)^2 + p_-^2\right)} \le m\tau^2 \left(1 - \frac{2(1-p_+)p_-}{(1-p_+)^2 + p_-^2}\right) \tag{27}$$

$$\le m\tau^2 \left(1 - \left(\frac{p_-}{p_+}\right)^2\right). \tag{28}$$

We can lower bound $p_-/p_+$ using Eq 17 and Eq 20 as:

$$\frac{p_-}{p_+} = \frac{1 + (K-2)e^{-\alpha\tau} + e^{-\alpha\tau/2}}{1 + (K-2)e^{-\alpha\tau/2} + e^{\alpha\tau/2}} \ge \frac{(K-2)e^{-\alpha\tau}}{3(K-2)e^{-\alpha\tau/2}} \ge \frac{e^{-\alpha\tau/2}}{3}, \tag{29}$$

where we use $\alpha\tau \le \log(K-2)$ to upper bound each of the denominator terms by $(K-2)e^{-\alpha\tau/2}$.

Plugging this back, we get the following bound on the quality of one-hot gradients:

$$\lim_{D \to \infty} \frac{\|\mathbf{u}_k^{\|}\|^2}{\|\mathbf{u}_k^{\perp}\|^2} \le m\tau^2 \left(1 - \frac{e^{-\alpha\tau}}{9}\right). \tag{30}$$

Now let $p_+^{\mathsf{te}}$ and $p_-^{\mathsf{te}}$ denote the probabilities of a *perfect* teacher. We can apply an identical argument for distillation to bound the ratio of the gradients as:

$$\lim_{D \to \infty} \frac{\|\tilde{\mathbf{u}}^{\|}\|^2}{\|\tilde{\mathbf{u}}^{\perp}\|^2} := \frac{m^2 \tau^2 ((p_+^{\mathsf{te}} - p_+) + (p_-^{\mathsf{te}} - p_-))}{m((p_+^{\mathsf{te}} - p_+)^2 + (p_-^{\mathsf{te}} - p_-)^2)} \ge m\tau^2 \left(1 + 2\frac{(p_+^{\mathsf{te}} - p_+)(p_-^{\mathsf{te}} - p_-)}{(p_+^{\mathsf{te}} - p_+)^2 + (p_-^{\mathsf{te}} - p_-)^2}\right) \tag{31}$$

$$\ge m\tau^2 \left(1 + 2\frac{\kappa^2 p_+^{\mathsf{te}} p_-^{\mathsf{te}}}{(p_+^{\mathsf{te}})^2 + (p_-^{\mathsf{te}})^2}\right) \ge m\tau^2 (1 + \kappa^2), \tag{32}$$

where in the second line we've used Lemma B.1 to lower-bound the teacher-student probability gap, subsequently followed by the AM-GM inequality. Here, we have used the fact that $p_-^{\mathsf{te}} - p_- > 0$, or in other words, that the gradients from $S_-$ are constructive.

$\square$

We are now ready to state and prove our main theorem:

**Theorem B.4.** *Consider a student model with weights $\mathbf{w}_k = \alpha \boldsymbol{\mu}_k^{\star}$ such that $\alpha \le \frac{1}{\tau} \log(K-2)$. Let $\mathbf{u}_k$ denote the one-hot gradients on this model under the cross-entropy loss on the dataset $S$.*

*Consider an imperfect teacher model with weights $\mathbf{w}_k$ such that $\|\mathbf{w}_k\| = \alpha^{te}$ and $\frac{\|\mathbf{w}_k^{\perp}\|^2}{\|\mathbf{w}_k^{\|}\|^2} \le \epsilon^{te}$ where $\epsilon^{te} > \frac{1}{m\tau^2}$. Assume the teacher model is early-stopped in that $\alpha^{te} \le \frac{1}{\tau} \log(K-2)$.*

*Assuming $\alpha^{te} \ge \frac{1}{\tau} \log 3$, let $\kappa > 0$ denote the level of early-stopping in the student, such that $\alpha^{te} - \alpha \ge \frac{2}{\tau} \log \frac{3}{1-\kappa}$. Let $\tilde{\mathbf{u}}_k$ denote the gradient under distillation loss with this teacher on the above student model.*

*Assume that the simplifying assumptions of Assumption B.1 hold. Then, for a sufficiently large $\tau$, there exists values of $\alpha, \kappa$ such that $\alpha \in [0, \alpha^{te}]$ such that $\kappa^3 \ge 192 \alpha^{te} \tau \sqrt{m\epsilon^{te}}$ and so we have for any class $k$ that:*

$$\underbrace{\lim_{D \to \infty} \frac{\|\mathbf{u}_k^{\|}\|^2}{\|\mathbf{u}_k^{\perp}\|^2} \le m\tau^2 \left(1 - \frac{e^{-\alpha\tau}}{9}\right)}_{\textit{Upper-bound on quality of one-hot gradients}} \le \underbrace{m\tau^2 \left(1 + \frac{\kappa^2}{2}\right) \le \lim_{D \to \infty} \frac{\|\tilde{\mathbf{u}}_k^{\|}\|^2}{\|\tilde{\mathbf{u}}_k^{\perp}\|^2}}_{\textit{Lower-bound on quality of distillation gradients}}. \tag{33}$$

*Furthermore, the quality of the teacher weights can be upper bounded by the quality of distillation gradients as :*

$$\frac{1}{\epsilon^{te}} \leq \underbrace{m\tau^2 \left(1 + \frac{\kappa^2}{2}\right)}_{\text{Lower-bound on quality of distillation gradients}} \leq \lim_{D\to\infty} \frac{\|\tilde{\mathbf{u}}_k^{\|}\|^2}{\|\tilde{\mathbf{u}}_k^{\perp}\|^2} . \tag{34}$$

*Proof.* We will now extend the proof of Lemma B.3 to an imperfect teacher with probabilities $\mathbf{q}^{te}(\mathbf{x})$. For convenience, let us assume that for $\mathbf{x} \in S_+$ $|p_k^{te}(\mathbf{x}) - q_k^{te}(\mathbf{x})| \leq \gamma_+$ and $\mathbf{x} \in S_-$, $|p_k^{te}(\mathbf{x}) - q_k^{te}(\mathbf{x})| \leq \gamma_-$ for some constants $\gamma_+$ and $\gamma_-$. We can say that:

$$\|\tilde{\mathbf{u}}^{\|}\|^2 = \tau^2 \left( \sum_{\mathbf{x}\in S_-\cup S_+} q_k^{te}(\mathbf{x}) - p_k(\mathbf{x}) \right)^2 \geq m^2\tau^2 \left( (p_+^{te} - p_+) + (p_-^{te} - p_-) - \gamma_+ - \gamma_- \right)^2 . \tag{35}$$

If we have $\gamma_- \leq \frac{\epsilon'}{2}\kappa p_-^{te}$ for some $\epsilon'$, from Lemma B.1, we have $\gamma_- \leq \frac{\epsilon'}{2}\kappa(p_-^{te} - p_-)$. We can make a similar claim for $\gamma_+$. Thus,

$$\|\tilde{\mathbf{u}}^{\|}\|^2 \geq m^2\tau^2 \left( (p_+^{te} - p_+) + (p_-^{te} - p_-) \right)^2 \left(1 - \frac{\epsilon'}{2}\right)^2 \tag{36}$$

$$\geq m^2\tau^2 \left( (p_+^{te} - p_+) + (p_-^{te} - p_-) \right)^2 (1 - \epsilon') \tag{37}$$

$$\tag{38}$$

As for the orthogonal gradient term, recall that $\|\mathbf{u}_k^{\perp}\|^2$ is essentially the summation of $D-1$ squared random variables. With a perfect teacher, we could write each of the $D-1$ variables as a sum of $m$ Gaussians each scaled by a constant probability $(p_+^{te} - p_+)$ and $(p_-^{te} - p_-)$ independent of the sample $\mathbf{x}$. However, with the imperfect teacher, the probabilities are themselves random variables dependent on the draw of $\mathbf{x}$. So we first bound the gap between this norm and the norm under the perfect teacher.

Without loss of generality, let us denote the $D-1$ dimensions of $\mathbf{x}_k$ that is orthogonal to $\boldsymbol{\mu}_k^{\star}$ as $x_{k,1}, x_{k,2}, \ldots, x_{k,D-1}$. Then,

$$\lim_{D\to\infty} \|\tilde{\mathbf{u}}^{\perp}\|^2 \leq \lim_{D\to\infty} \sum_{j=1}^{D-1} \left( \sum_{\mathbf{x}\in S_+\cup S_-} (p_k^{te}(\mathbf{x}) - p_k(\mathbf{x}))x_{k,j} \right)^2 \tag{39}$$

$$\leq \lim_{D\to\infty} \sum_{j=1}^{D-1} \left( \sum_{\mathbf{x}\in S_+\cup S_-} (q_k^{te}(\mathbf{x}) - p_k(\mathbf{x}))x_{k,j} + x_{k,j} \underbrace{(p_k^{te}(\mathbf{x}) - q_k^{te}(\mathbf{x}))}_{\leq \gamma_+ \text{ or } \leq \gamma_-} \right)^2 \tag{40}$$

$$\leq \underbrace{\lim_{D\to\infty} \sum_{j=1}^{D-1} \left( \sum_{\mathbf{x}\in S_+\cup S_-} (p_k^{te}(\mathbf{x}) - p_k(\mathbf{x}))x_{k,j} \right)^2}_{\leq m((p_+^{te}-p_+)^2 + (p_-^{te}-p_-)^2)} \tag{41}$$

$$+ \underbrace{\lim_{D\to\infty} \sum_{j=1}^{D-1} \left( \sum_{\mathbf{x}\in S_+} \gamma_+ |x_{k,j}| + \sum_{\mathbf{x}\cup S_-} \gamma_- |x_{k,j}| \right)^2}_{A_j} \tag{42}$$

$$+ \lim_{D\to\infty} 2 \sum_{j=1}^{D-1} \underbrace{\sum_{\mathbf{x}\in S_+\cup S_-} (p_k^{\text{te}}(\mathbf{x}) - p_k(\mathbf{x}))|x_{k,j}| \left( \sum_{\mathbf{x}\in S_+} \gamma_+|x_{k,j}| + \sum_{\mathbf{x}\in S_-} \gamma_-|x_{k,j}| \right)}_{B_j}$$

$$(43)$$

$$(44)$$

For each $j$, $A_j$ is the squared of the sum of the absolute value $m$ Gaussians with variance $\frac{\gamma_+^2}{D}$ and another $m$ with variance $\frac{\gamma_-^2}{D}$. The means of these variables are $\gamma_+\sqrt{\frac{1}{D}\frac{2}{\pi}}$ and $\gamma_-\sqrt{\frac{1}{D}\frac{2}{\pi}}$ respectively. The expectation of $A_j$ is the squared sum of the means and the sum of the variances of all these variables, equalling $\frac{m(\gamma_+^2+\gamma_-^2)}{D} + m^2\left(\gamma_+\sqrt{\frac{2}{D}\frac{2}{\pi}} + \gamma_-\sqrt{\frac{1}{D}\frac{2}{\pi}}\right)^2$. The second term can be upper bounded by $m^2\left(\frac{\gamma_+^2}{D} + \frac{\gamma_-^2}{D}\right)$. Therefore, we can conclude that $\lim_{D\to\infty}\sum_{j=1}^{D-1} A_j \leq 3m^2(\gamma_+^2+\gamma_-^2)$.

For each $j$, $B_j$ can be expanded into $(2m)^2$ terms. The expected value of each of these terms can be bounded in the form of $(p_k^{\text{te}}(\mathbf{x}) - p_k(\mathbf{x}))\left(\frac{\gamma_+}{D}\right)$ or $(p_k^{\text{te}}(\mathbf{x}) - p_k(\mathbf{x}))\left(\frac{\gamma_-}{D}\right)$. Thus, the expected value of $B_j$ is $m\left(\frac{\gamma_-}{D} + \frac{\gamma_+}{D}\right)\sum_{\mathbf{x}}(p_k^{\text{te}}(\mathbf{x}) - p_k(\mathbf{x}))$. Let $\lim_{D\to\infty} B_j = m(\gamma_- + \gamma_+)\sum_{\mathbf{x}}(p_k^{\text{te}}(\mathbf{x}) - p_k(\mathbf{x}))$

Now, if we have that $\sqrt{m}\gamma_- \leq \frac{\epsilon'}{4}\kappa p_-^{\text{te}}$ and $\sqrt{m}\gamma_+ \leq \frac{\epsilon'}{4}\kappa p_+^{\text{te}}$, for some $\epsilon'$ we have:

$$\lim_{D\to\infty}\|\tilde{\mathbf{u}}^\perp\|^2 \leq m((p_+^{\text{te}} - p_+)^2 + (p_-^{\text{te}} - p_-)^2) \tag{45}$$

$$+ 3m\frac{(\epsilon')^2}{16}\left((\kappa p_+^{\text{te}})^2 + (\kappa p_-^{\text{te}})^2\right) + m\frac{\epsilon'}{4}(\kappa p_+^{\text{te}} + \kappa p_-^{\text{te}})(p_+^{\text{te}} - p_+) + (p_-^{\text{te}} - p_-)) \tag{46}$$

$$(47)$$

In the second term, we can upper bound $(\kappa p_+^{\text{te}})^2 + (\kappa p_-^{\text{te}})^2$ by $((p_+^{\text{te}} - p_+)^2 + (p_-^{\text{te}} - p_-)^2)$ using Lemma B.1. In the third term, we can similarly upper bound $\kappa p_+^{\text{te}}$ and $\kappa p_-^{\text{te}}$ followed by an AM-GM inequality to upper-bound by $2((p_+^{\text{te}} - p_+)^2 + (p_-^{\text{te}} - p_-)^2)$. Therefore,

$$\lim_{D\to\infty}\|\tilde{\mathbf{u}}^\perp\|^2 \leq m((p_+^{\text{te}} - p_+)^2 + (p_-^{\text{te}} - p_-)^2)\left(1 + 3\frac{(\epsilon')^2}{16} + \frac{\epsilon'}{4}\right) \tag{48}$$

$$\leq m((p_+^{\text{te}} - p_+)^2 + (p_-^{\text{te}} - p_-)^2)(1 + \epsilon'). \tag{49}$$

$$(50)$$

Then, we have that for our imperfect teacher:

$$\lim_{D\to\infty}\frac{\|\tilde{\mathbf{u}}^\|\|^2}{\|\tilde{\mathbf{u}}^\perp\|^2} \geq m\tau^2(1+\kappa^2)\frac{(1-\epsilon')}{(1+\epsilon')} \geq m(1+\kappa^2)(1-\epsilon')^2$$

$$\geq m\tau^2(1+\kappa^2)(1-2\epsilon') \geq m\tau^2(1+\kappa^2-4\epsilon') \geq m\left(1+\frac{\kappa^2}{2}\right).$$

The last step follows if we have $\epsilon' \leq \frac{\kappa^2}{8}$. Plugging this back in our required inequality of the form $\sqrt{m}\gamma_- \leq \frac{\epsilon'}{4}\kappa p_-^{\text{te}}$, we need $\sqrt{m}\gamma_- \leq \frac{\kappa^3}{16}p_-^{\text{te}}$ and similarly for $\gamma_+$. From Lemma B.2, this means we need $\kappa^3 \geq 192\alpha^{\text{te}}\tau\sqrt{m\epsilon^{\text{te}}}$.

Now, we need to make sure that there are possible values of $\alpha$ for which the above lower-bound on $\kappa$ can be realized. Note that $\kappa$ can only attain values of $1 - 3e^{-(\alpha^{\text{te}}-\alpha)\frac{\tau}{2}}$ for various values of $\alpha$. This follows from the statement of Lemma B.1. Thus, we want $1 - 3e^{-(\alpha^{\text{te}}-\alpha)\frac{\tau}{2}} > \left(192\alpha^{\text{te}}\tau\sqrt{m\epsilon^{\text{te}}}\right)$ for there to be any values of $\alpha$ to achieve our lower-bound on $\kappa$. Furthermore, we have that $\epsilon = \frac{c}{m\tau^2}$ for some $c > 1$. Thus, we need $1 - 3e^{-(\alpha^{\text{te}}-\alpha)\frac{\tau}{2}} > \left(192\alpha^{\text{te}}\sqrt{c}\right)$ to have feasible solutions for $\alpha$. This is possible as long as $\tau$ is sufficiently large. $\qquad\square$

| Parameter | CIFAR10* | Tiny-ImageNet |
|---|---|---|
| Weight decay | $10^{-4}$ | $5 \cdot 10^{-4}$ |
| Batch size | 1024 | 128 |
| Epochs | 450 | 200 |
| Peak learning rate | 1.0 | 0.1 |
| Learning rate warmup epochs | 15 | 5 |
| Learning rate decay factor | 0.1 | 0.1 |
| Learning rate decay epochs | $200, 300, 400$ | $75, 135$ |
| Nesterov momentum | 0.9 | 0.9 |
| Distillation weight | 1.0 | 1.0 |
| Distillation temperature | 4.0 | 4.0 |
| Gradual loss switch window | $1k$ steps | $10k$ steps |

Table 1: Summary of training settings.

## C EXPERIMENTS AND PLOTS

### C.1 DETAILS OF EXPERIMENTAL SETUP

We present details on relevant hyper-parameters for our experiments.

**Model architectures**. For all image datasets (CIFAR10, CIFAR100, Tiny-ImageNet), we use ResNet-v2 (He et al., 2016a) and MobileNet-v2 (Sandler et al., 2018), models. Specifically, for CIFAR, we consider the CIFAR ResNet-$\{56, 20\}$ family and MobileNet-v2 architectures; for Tiny-ImageNet, we consider the ResNet-$\{50, 18\}$ family and MobileNet-v2 architectures; For all ResNet models, we employ standard augmentations as per He et al. (2016b).

For all text datasets (MNLI, AGNews, QQP, IMDB), we fine-tune a pre-trained RoBERTa (Liu et al., 2019) model. We consider a combinations of cross-architecture- and self-distillation with RoBERTa -Base, -Medium and -Small architectures.

**Training settings**. We train using minibatch SGD applied to the softmax cross-entropy loss. For all image datasets, we follow the settings in Table 1.

For all text datasets, we use a batch size of 64, and train for 25000 steps. We use a peak learning rate of $10^{-5}$, with 1000 warmup steps, decayed linearly. For the distillation experiments, we use a distillation weight of 1.0. We use temperature $\tau = 2.0$ for MNLI, $\tau = 16.0$ for IMDB, $\tau = 1.0$ for QQP, and $\tau = 1.0$ for AGNews.

### C.2 ADDITIONAL RESULTS

#### C.2.1 SCATTER PLOTS OF PROBABILITIES

In this section, we present additional scatter plots of the teacher-student logit-transformed probabilities for the class corresponding to the teacher's top prediction: Fig 5 (for CIFAR100), Fig 6 (for TinyImagenet), Fig 7 (for CIFAR10), Fig 8 (for MNLI and AGNews), Fig 9 (for self-distillation on QQP, IMDB and AGNews) and Fig 10 (for cross-architecture distillation on language datasets). We find underfitting of the hard points in a majority of the cases, with a few **notable exceptions and caveats**:

1. For MobileNet self-distillation on CIFAR100, and for a majority of the CIFAR10 experiments, we find no underfitting of the harder points *on the training dataset*. However, we do find underfitting of the harder points *on the test dataset*. Interestingly, we also find an underfitting of easier points in the training dataset.

2. In the language datasets, we generally find the plots to be different in pattern from the image datasets. In particular we find that, for harder points, there is both significant underfitting

| Dataset | Teacher | Student | Train accuracy | | | Test accuracy | | |
|---|---|---|---|---|---|---|---|---|
| | | | Teacher | Student (OH) | Student (DIST) | Teacher | Student (OH) | Student (DIST) |
| CIFAR10 | ResNet-56 | ResNet-56 | 100.00 | 100.00 | 100.00 | 93.72 | 93.72 | 93.9 |
| | ResNet-56 | ResNet-20 | 100.00 | 99.95 | 99.60 | 93.72 | 91.83 | 92.94 |
| | ResNet-56 | MobileNet-v2-1.0 | 100.00 | 100.00 | 99.96 | 93.72 | 85.11 | 87.81 |
| | MobileNet-v2-1.0 | MobileNet-v2-1.0 | 100.00 | 100.00 | 100.00 | 85.11 | 85.11 | 86.76 |
| CIFAR100 | ResNet-56 | ResNet-56 | 99.97 | 99.97 | 97.01 | 72.52 | 72.52 | 74.55 |
| | ResNet-56 | ResNet-20 | 99.97 | 94.31 | 84.48 | 72.52 | 67.52 | 70.87 |
| | MobileNet-v2-1.0 | MobileNet-v2-1.0 | 99.97 | 99.97 | 99.96 | 54.32 | 54.32 | 56.32 |
| | ResNet-56 | MobileNet-v2-1.0 | 99.97 | 99.97 | 99.56 | 72.52 | 54.32 | 62.4 |
| CIFAR100 noise | ResNet-56 | ResNet-56 | 99.9 | 99.9 | 95.6 | 69.8 | 69.8 | 72.7 |
| | ResNet-56 | ResNet-20 | 99.9 | 91.4 | 82.8 | 69.8 | 64.9 | 69.2 |
| Tiny-ImageNet | ResNet-50 | ResNet-50 | 98.62 | 98.62 | 94.84 | 66 | 66 | 66.44 |
| | ResNet-50 | ResNet-18 | 98.62 | 93.51 | 91.09 | 66 | 62.78 | 63.98 |
| | ResNet-50 | MobileNet-v2-1.0 | 98.62 | 89.34 | 87.90 | 66 | 62.75 | 63.97 |
| | MobileNet-v2-1.0 | MobileNet-v2-1.0 | 89.34 | 89.34 | 82.26 | 62.75 | 62.75 | 63.28 |
| MNLI | RoBERTa-Base | RoBERTa-Small | 92.9 | 72.1 | 72.6 | 87.4 | 69.9 | 70.3 |
| MNLI | RoBERTa-Small | RoBERTa-Small | 72.1 | 72.1 | 71.0 | 69.9 | 69.9 | 69.9 |
| MNLI | RoBERTa-Medium | RoBERTa-Medium | 88.2 | 88.2 | 85.6 | 83.8 | 83.8 | 83.5 |
| IMDB | RoBERTa-Small | RoBERTa-Small | 100.0 | 100.0 | 99.1 | 90.4 | 90.4 | 91.0 |
| QQP | RoBERTa-Small | RoBERTa-Small | 85.0 | 85.0 | 83.2 | 83.5 | 83.5 | 82.5 |
| QQP | RoBERTa-Medium | RoBERTa-Medium | 92.3 | 92.3 | 90.5 | 89.7 | 89.7 | 89.0 |
| AGNews | RoBERTa-Small | RoBERTa-Small | 96.5 | 96.5 | 95.9 | 93.8 | 93.8 | 93.9 |
| AGNews | RoBERTa-Medium | RoBERTa-Medium | 98.6 | 98.6 | 97.9 | 94.6 | 94.6 | 94.6 |

Table 2: Summary of train and test performance of various distillation settings.

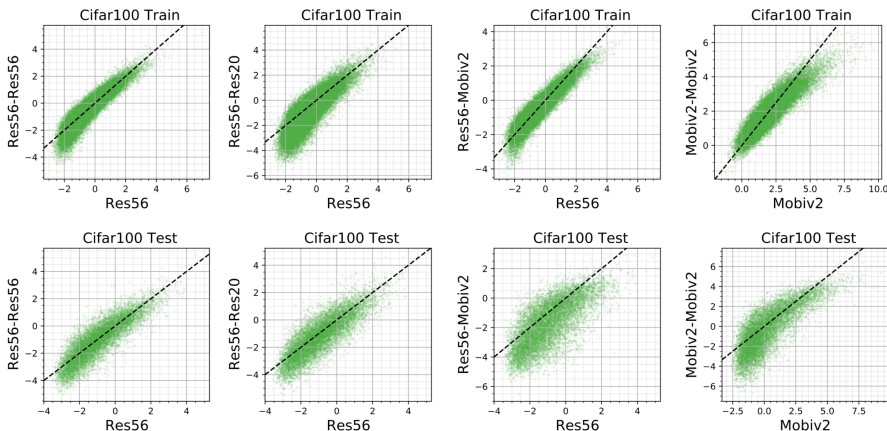

Figure 5: **Teacher-student logit plots for CIFAR100 experiments:** We report plots for various distillation settings involving ResNet56, ResNet20 and MobileNet-v2. We find underfitting of the hard points in the training set in all but the MobileNet self-distillation setting. Nevertheless, even in the MobileNet self-distillation setting, we find significant underfitting in the *test* dataset.

and overfitting. For easier points, there is less deviation, and if any, the deviation is from overfitting. We interpret this as the regularization from distillation inducing *less precision* on the harder points, rather than an underfitting per se.

3. Our patterns generally break down in the *cross-architecture* settings of language datasets (with the exception of a couple of settings). We suspect that this may be because certain cross-architecture effects dominate over the underfitting, which is a much more subtle effect.

4. The underfitting we observe in the language datasets are not always associated with an improvement in the student's generalization. e.g., we find no improvement in AGNews, and a decrease in generalization in QQP (see Table 2). (Note that we didn't finetune hyperparameters in these settings to make distillation work.)

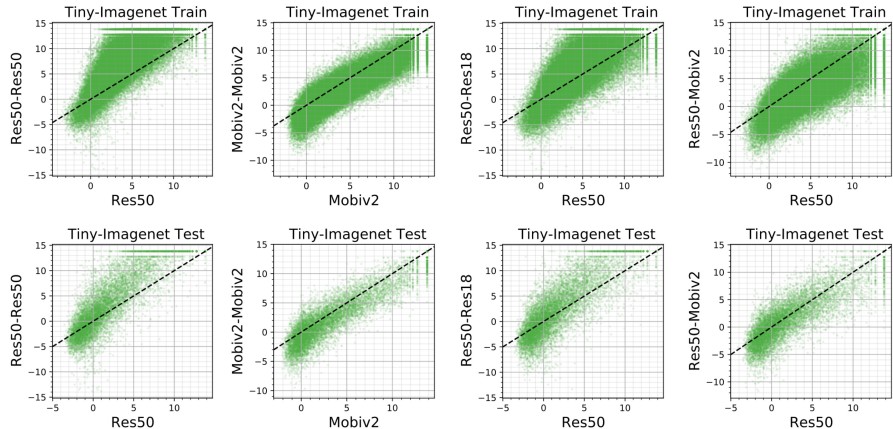

Figure 6: **Teacher-student logit plots for Tiny-Imagenet experiments:** We report plots for various distillation settings involving ResNet50, ResNet18 and MobileNet-v2. We find underfitting of the hard points in all the settings. We also find *overfitting of the easier points* when the student is a ResNet.

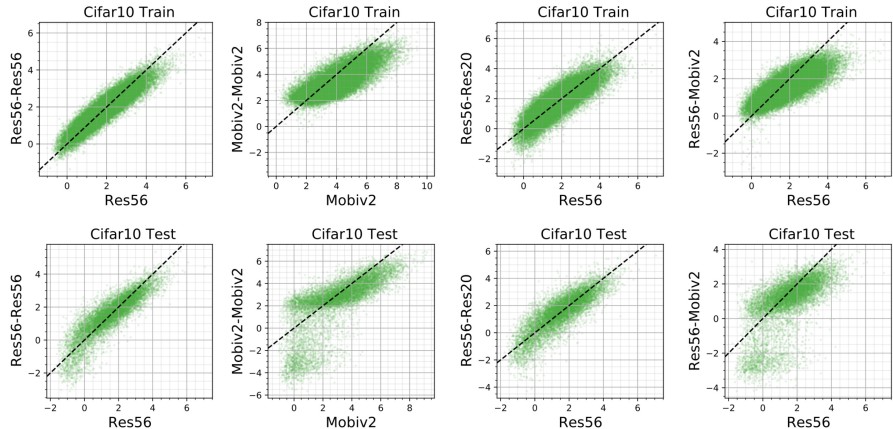

Figure 7: **Teacher-student logit plots for CIFAR10 experiments:** We report plots for various distillation settings involving ResNet56, ResNet20 and MobileNet-v2. We find that the underfitting phenomenon is almost non-existent in the training set (except for ResNet50 to ResNet20 distillation). However the phenomenon is prominent in the test dataset.

### C.2.2 TEACHER'S PREDICTED CLASS VS. GROUND TRUTH CLASS

Recall that in all our scatter plots we have looked at the probabilities of the teacher and the student on the teacher's predicted class i.e., $(p_{y^{te}}^{te}(x), p_{y^{te}}^{st}(x))$ where $y^{te} \doteq \mathrm{argmax}_{y' \in [K]} \ p_{y'}^{te}(x)$. Another natural alternative would have been to look at the probabilities for the ground truth class, $(p_{y^\star}^{te}(x), p_{y^\star}^{st}(x))$ where $y^\star$ is the ground truth label. We chose to look at $y^{te}$ however, because we are interested in the "shortcomings" of the distillation procedure where the student only has access to teacher probabilities and not ground truth labels.

Nevertheless, one may still be curious as to what the probabilities for the ground truth class look like. First, we note that the plots look almost identical for the *training dataset* owing to the fact that the teacher model typically fits the data to zero training error (we skip these plots to avoid redundancy). However, we find stark differences in the test dataset as shown in Fig 11. In particular, we see that the underfitting phenomenon is no longer prominent, and almost non-existent in many of our settings.

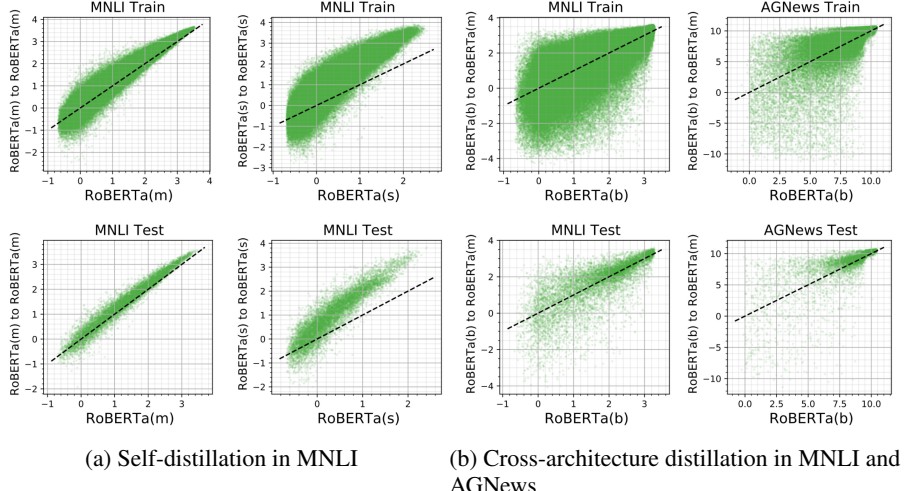

(a) Self-distillation in MNLI
(b) Cross-architecture distillation in MNLI and AGNews

Figure 8: **Teacher-student logit plots for MNLI and AGNews experiments:** We report plots for various distillation settings involving RoBERTa models. On the **left**, in the self-distillation settings on MNLI, we find significant underfitting of hard points (and also overfitting), while easy points are completely overfit. On the **right**, we report cross-architecture ( Base to Medium) distillation for MNLI and AGNews. Here the plots are not as typical of our other plots. Nevertheless, we observe significant underfitting *and* overfitting of hard points, and overfitting of the extremely easy points. We interpret this as distillation reducing its "precision" on the harder points (perhaps by ignoring lower eigenvectors that provide finer precision).

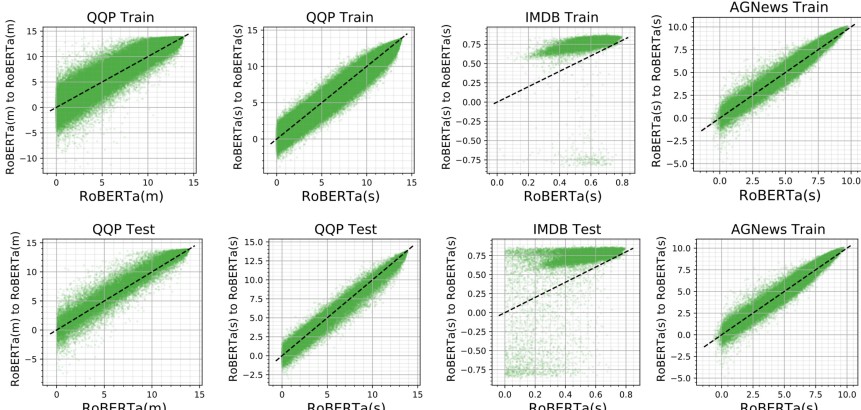

Figure 9: **Teacher-student logit plots for self-distillation in language datasets (QQP, IMDB, AGNews):** We report plots for various self-distillation settings involving RoBERTa models. As in the other language dataset settings, we find both significant underfitting and overfitting for harder points (indicating lack of precision), and with more precision for easier points (typically with more overfitting).

This is surprising as this suggests that the student somehow matches the probabilities on the ground truth class despite not knowing what the ground truth class is. This also tells us the mechanism by which the student manages to correct some of the teacher's labels: by reducing the probability of non-ground-truth classes rather than increasing the probability of the ground truth class. Indeed, in Fig 12, we verify this to be the case. We dissect the test data plots into four parts depending on which of the teacher and student model gets the label right. We consistently find that the points that the student gets right but the teacher not, fall in the underfit set of points.

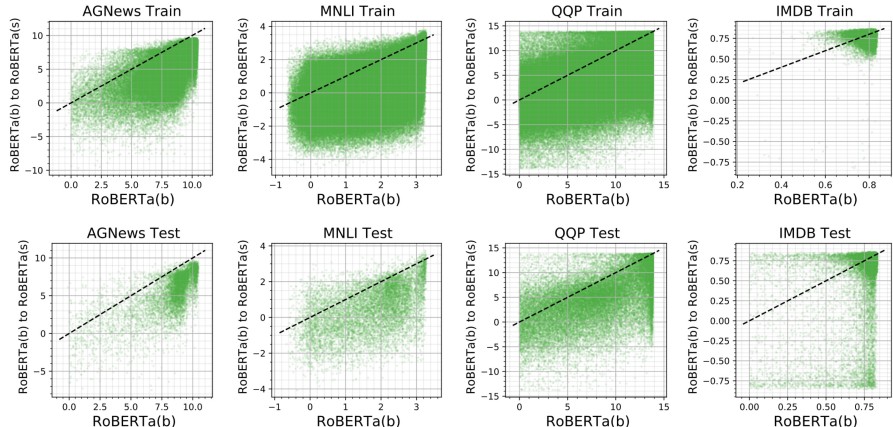

Figure 10: **Teacher-student logit plots for cross-architecture distillation in language datasets (AGNews, QQP, IMDB, MNLI):** We report plots for various cross-architecture distillation settings involving RoBERTa models. While we find significant student-teacher deviations in these settings, our typical patterns do not apply here. We believe that effects due to "cross-architecture gaps" may have likely drowned out the underfitting patterns, which is a more subtle phenomenon that shines in self-distillation settings.

Interestingly, the underfit set of points is roughly *equivalent* to the set of all points where *at least one of the models is incorrect* This suggests that in its attempt to underfit some of the points, the student can get some incorrect, potentially points which are inherently fuzzy (e.g., they are similar to multiple classes). Theorem 4.2 suggests that the student would use these points to improve its features, thereby increasing accuracy on other points that are not inherently as fuzzy.

Finally, we note that previous work (Lukasik et al., 2021) has examined deviations on ground truth class probabilities albeit in an aggregated sense (at a class-level rather than at a sample-level). While they find that the student tends to have lower ground truth probability than the teacher on problems with label imbalance, they do *not* find any such difference on standard datasets without imbalance. This is in alignment with what we find above.

### C.2.3    THE EFFECT OF LOSS SWITCHING

We provide additional results on loss-switching in this section. Fig 14 presents the accuracy values under a one-hot-to-distillation switch. Fig 15 presents the accuracy values under a distillation-to-one-hot switch. Fig 13 presents student-teacher scatter plots over the course of training under all the four methods (with and without loss switches). Fig 16 presents the trajectory of accuracies for ResNet50 on TinyImagenet under all the four settings.

It is worth noting a nuance in this story. First, switching to one-hot typically preserves the gains of distillation if it is not run for too long — and in the rare case of ResNet50 self-distillation on TinyImagenet even outperforms distillation — which supports earlier findings (Cho & Hariharan, 2019; Zhou et al., 2020; Jafari et al., 2021) that advocate a soft final switch to one-hot training. Overall, this seems to suggest that, the early phase of distillation may be a *sufficient* requirement to gain all the benefits of distillation (even if our earlier experiments suggest that they are not *necessary*). *However*, we consistently find that switching to and training with one-hot for a sufficiently long time deteriorates the gains made by distillation (Fig 1b, (Fig 16).

### C.3    ABLATIONS

Here, we conduct two experiments showing that the underfitting phenomenon holds under other conditions, specifically (a) longer training of the student and (b) smaller batch sizes and learning rate.

**Longer training:** In Fig 17 (left two images), we conduct experiments where we run knowledge distillation with the ResNet-56 student on CIFAR100 for $2.3\times$ longer ($50k$ steps instead of $21.6k$

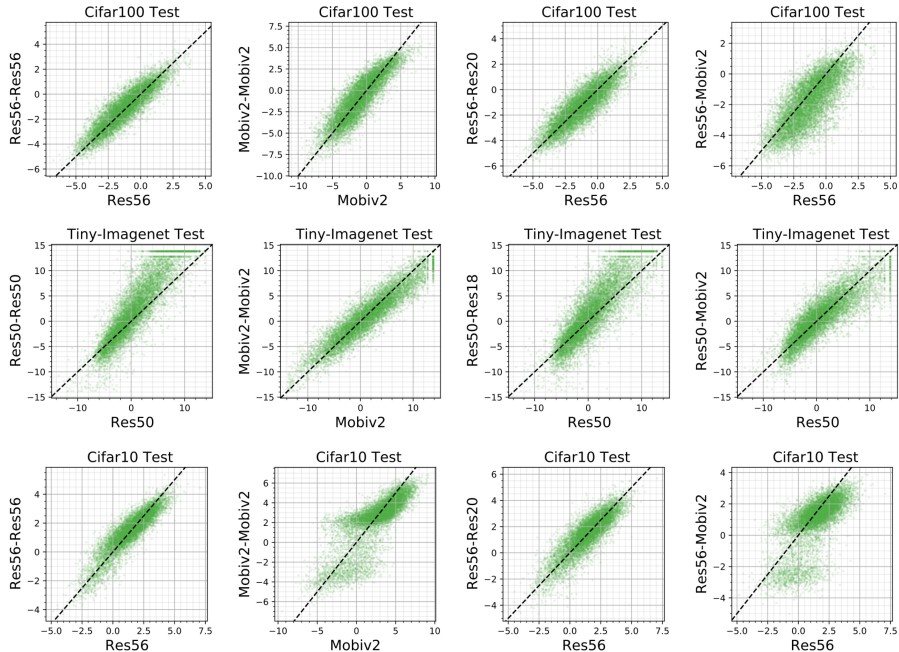

Figure 11: **Scatter plots for ground truth class:** Unlike in other plots where we report the probabilities for the class predicted by the teacher, here we focus on the ground truth class. Recall that the $X$-axis corresponds to the teacher, the $Y$-axis to the student, and all the probabilities are log-transformed. Surprisingly, we observe a much more subdued underfitting here, with the phenomenon completely disappearing in many situations. This suggests that the student magically *preserves* the ground-truth probabilities *despite no knowledge of what the ground-truth class is*, while underfitting on the teacher's predicted class.

steps overall) and with the ResNet-50 student on TinyImagenet for about $2\times$ longer ($300k$ steps over instead of roughly $150k$ steps). We find the resulting plots to continue to have the same underfitting as the earlier plots. It is worth noting that in contrast, in a linear setting, it is reasonable to expect the underfitting to disappear after sufficiently long training. Therefore, the persistent underfitting in the non-linear setting is remarkable and suggests one of two possibilities:

- The underfitting is persistent simply because the student is not trained sufficiently long enough i.e., perhaps, when trained $10\times$ longer, the network might end up fitting the teacher probabilities perfectly.

- The network has reached a local optimum of the knowledge distillation loss and can never fit the teacher perfectly. This may suggest an added regularization effect in distillation, besides the eigenspace regularization that introduces the hard-point-underfitting in the first place. This unknown regularization effect may perhaps disappear if the learning rate was even smaller or if the student network was larger than the teacher network, allowing to reach higher precision.

**Smaller batch size/learning rate:** Finally, in Fig 17 (right image), we also verify that in the CIFAR100 setting if we set peak learning rate to $0.1$ (rather than $1.0$) and batch size to $128$ (rather than $1024$), our observations still hold.

### C.3.1   SCATTER PLOT FOR OTHER METRICS

In the main paper, recall that we look at student-teacher deviations via scatter plots of the probabilities of either models on the teacher's top class, *after applying a logit transformation*. It is natural to ask what these plots would look like under other variations. We explore this in Fig 18 for the CIFAR100 ResNet-56 self-distillation setting.

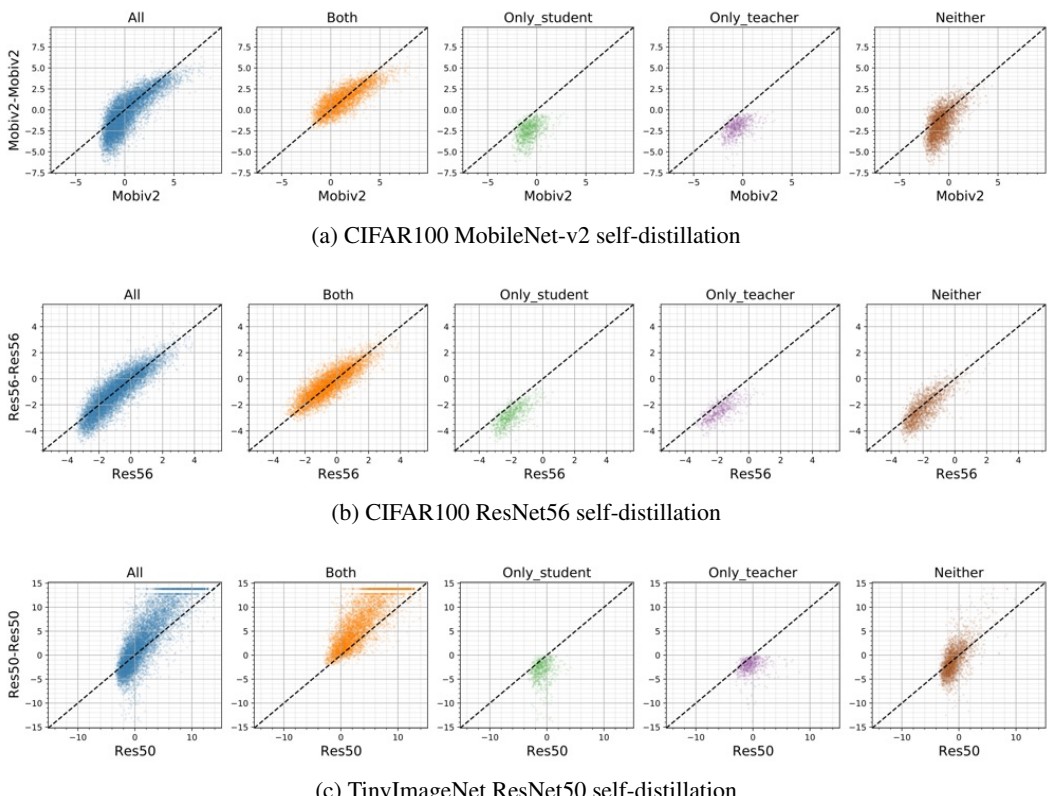

(a) CIFAR100 MobileNet-v2 self-distillation

(b) CIFAR100 ResNet56 self-distillation

(c) TinyImageNet ResNet50 self-distillation

Figure 12: **Dissecting the underfit points:** Across a few settings on TinyImagenet and CIFAR100, we separate the teacher-student scatter plots of logit-transformed probabilities into four subsets: subsets where both models's top prediction is correct (titled as Both), where only the student gets correct (Only_student), where only the teacher gets correct (Only_teacher), where neither get correct (Neither). We consistently find that the student's "underfit" points are points where at least one of the models go wrong.

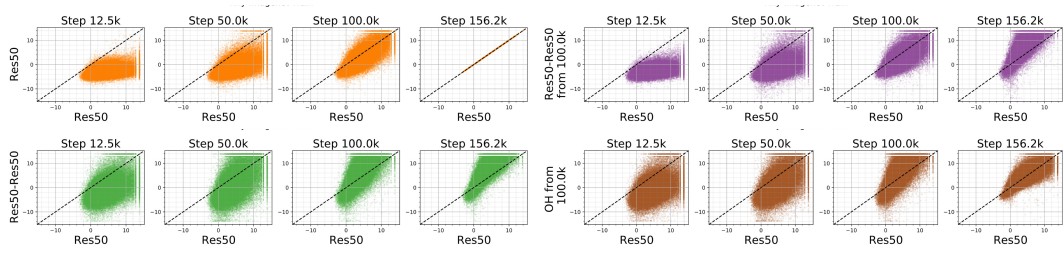

(a) One-hot and self-distillation.   (b) Loss-switching to distillation/one-hot at $100k$ steps.

Figure 13: **Evolution of logit-logit plots over various steps of training for TinyImageNet ResNet50 self-distillation setup:** On the **left**, we present plots for one-hot training (**top**) and distillation (**bottom**). Like in the case of CIFAR100, we again see a significant deviation between the standalone and student plots early on during training. At step 12.5k, we see that the standalone model has prioritized fitting some of the harder points close to $x = y$ line. The student has however fit easier points more substantially. On the **right**, we present similar plots for experiments from Section 3.2, with the loss switched to distillation (**top**) and one-hot (**bottom**) at $100k$ steps. From the last two visualized plots in each, observe that switching to distillation introduces (a) underfitting of hard points and overfitting of easier points, (b) while switching to one-hot curiously undoes both of this.

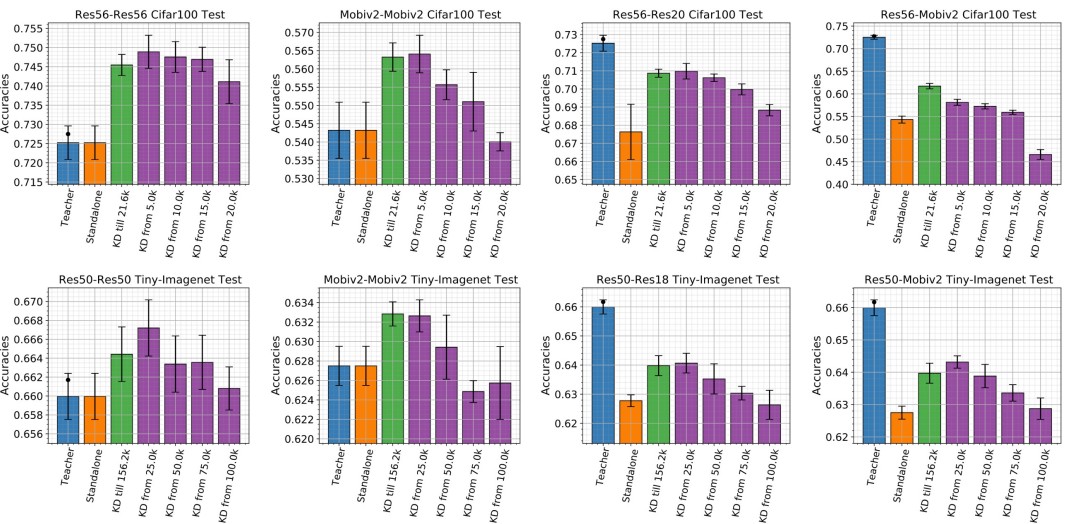

Figure 14: **Accuracies under late distillation for CIFAR100 and TinyImageNet:** We report the final accuracies for various distillation settings under ResNet56, ResNet20 and MobileNet-v2, with the loss switched to distillation in the middle of training. Observe that replacing the initial fourth/fifth of the training has zero effect on the distillation loss suggesting that deviations in the initial phase of training is not a crucial factor behind the success of distillation. Note that for MobileNet architectures, the loss-switch caused instability, which explains the strong dip in accuracy, even going below the one-hot model (top right).

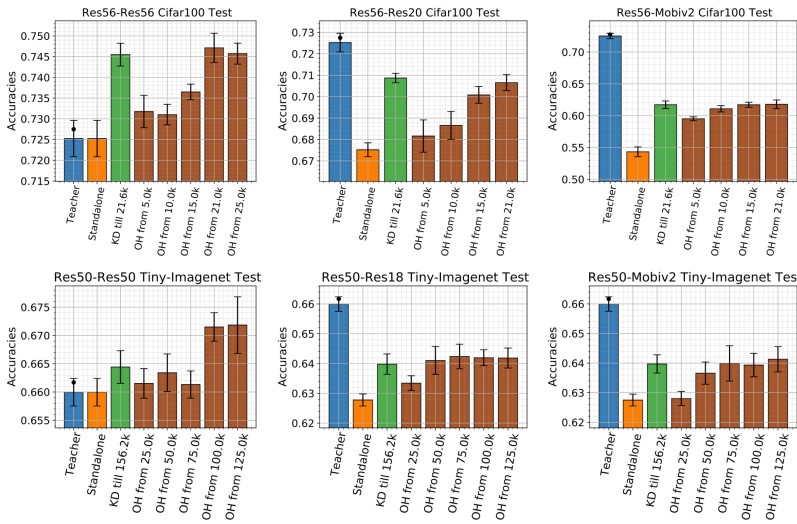

Figure 15: **Accuracies under late one-hot for CIFAR100 and TinyImageNet:** We report the final accuracies for various distillation-to-one-hot settings under ResNet56, ResNet20 and MobileNet-v2, with the loss switched to one-hot in the middle of training. Here, surprisingly, we find that in a majority of cases, switching to one-hot preserves the gains of distillation, and may even result an increase in accuracy, echoing the findings of (Zhang & Sabuncu, 2020; Yuan et al., 2020; Tang et al., 2020). However, in Fig 16, we find that these gains are subsequently destroyed by a longer one-hot training.

For quick reference, in the top row of Fig 18, we first show the standard logit-transformed probabilities plot where we find the underfitting phenomenon. In the second figure, we then directly plot the probabilities instead of applying the logit transformation on top of it. We find that the underfitting

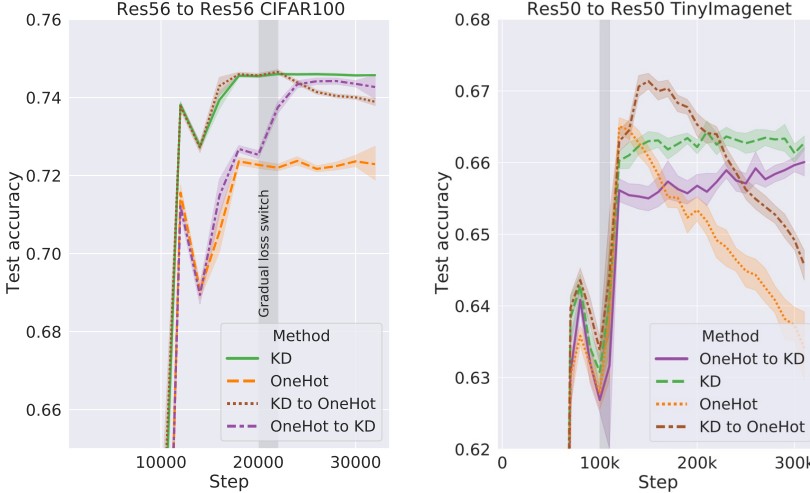

Figure 16: **Trajectory of test accuracy for loss-switching over longer periods of time:** We gradually change the loss for our self-distillation settings in CIFAR100 and TinyImagenet and extend training for a longer period of time. We find that in all cases, while switching to distillation gains in accuracy, switching to one-hot actively deteriorates the accuracy gains made by knowledge distillation. However, we note that for the Tiny-Imagenet, there is a small window of time for which there is a significant increase in accuracy under a switch to one-hot – this is later destroyed with longer training. Nevertheless, the overall finding here reinforces the intuition that the one-hot loss results in destructive gradients as discussed in Section 4.2.

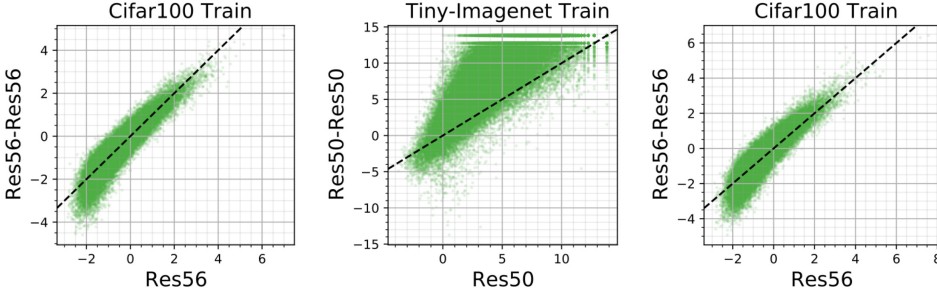

Figure 17: **Underfitting holds for longer runs and for smaller batch sizes:** For the self-distillation setting in CIFAR100 and TinyImagenet **(left two figures)**, we find that the student underfits teacher's hard points even after an extended period of training (roughly $2\times$ longer). On the **right**, we find in the CIFAR100 setting that underfitting occurs even for smaller batch sizes.

phenomenon does not prominently stand out here (although visible upon scrutiny, if we examine below the $X = Y$ line for $X \approx 0$). This illegibility is because small probability values tend to concentrate around 0; the logit transform however helps magnify the behavior of small probability values. For the third plot, we provide a scatter plot of entropy values of the teacher and student probability values to determine if the student distinctively deviates in terms of entropy from the teacher. It is not clear what characteristic behavior appears in this plot.

In the bottom plots, on the $Y$ axis we plot the KL-divergence of the student's probability from the teacher's probability. Along the $X$ axis we plot the same quantities as in the top row's three plots. Here, we observe interesting behavior across the board: the KL-divergence of the student tends to be higher on teacher's harder points, where "hard points" can be interpreted as either points where its top probability is low, or points where the teacher is "confused" enough to have high entropy.

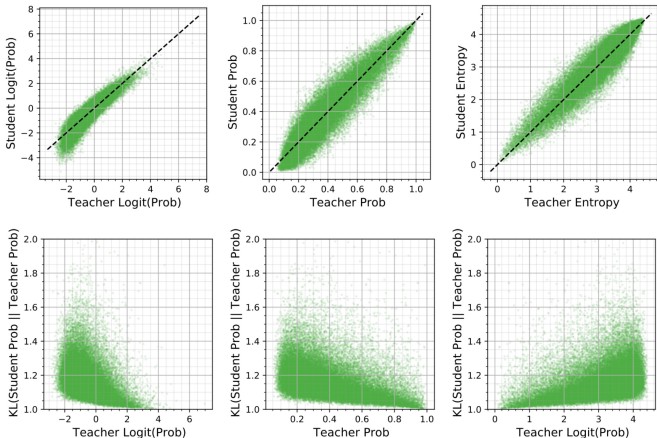

Figure 18: **Scatter plots for various metrics:** While in the main paper we presented scatter plots of logit-transformed probabilities, here we present scatter plots for various metrics, including the probabilities themselves, entropy of the probabilities, and the KL divergence of the student probabilities from the teacher. We find that the KL-divergence plots capture similar intuition as our logit-transformed probability plots. On the other hand, directly plotting the probabilities themselves is not as visually informative.

### C.4 VERIFYING THE EIGENSPACE VIEW EMPIRICALLY

In this section, we demonstrate the eigenspace view from Sec 4.1 in practice even in situations where our theoretical assumptions do not hold good. We go beyond our theoretical assumptions in the following ways:

1. We consider two self-distillation settings: first, a linear random features model trained on a noisy version of the MNIST dataset and next a 2-layer multi-layer perceptron (MLP) model trained on a subset of the MNIST dataset.

2. Both are trained with the cross-entropy loss (and not the squared error loss as used in our theory).

3. We consider a multi-class problem and so the output is not a single scalar value.

4. We use a finite learning rate with minibatches and Adam.

We provide exact details of these two settings at the end of the section

**Observations.** In short, we first observe in Fig 19 that in both these settings, the harder points of the teacher are underfit as usual. At the same time, we observe in Fig 20 and Fig 21 that the convergence rate of the student is much faster along the top eigendirections, when compared to the teacher (explained shortly). We show train-test accuracy plots in Fig 22.

To verify our eigenspace view theory, we show 2D projections of the trajectory of the teacher and student along two eigendirections picked at random (with the higher eigendirection one plotted along the $X$ axis). The final solution found is marked by a '$\star$', (typically at the top-right of the plot).

Here, we observe that the teacher already has an implicit bias towards converging faster along the top eigendirection (as is well-known). This can be inferred from the fact that the trajectories move quickly along the $X$ axis towards its final $X$ axis value, before making progress along the $Y$ axis. [1] But more interestingly, we find that for the student, the bias in this trajectory is more exaggerated; the student converges faster towards the final $X$ value of the teacher than the rate at which teacher gets there. In doing so, *the student covers a completely different part of parameter space never traversed*

---

[1]Intuitively, when this bias is extreme, the trajectory would reach its final $X$ axis value first with no displacement along the $Y$ axis, and only then progress along the $Y$ axis. Instead, we see a softer form of this bias, where the trajectory takes a "convex" shape.

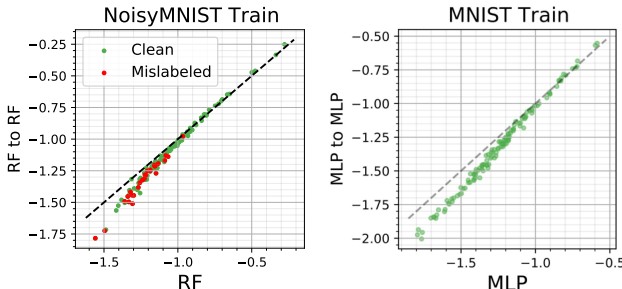

Figure 19: **Probability scatter plots verifying the eigenspace theory**: We observe the underfitting of hard points. The left plot above corresponds to the linear random features model trained on a noisy MNIST dataset, and the right corresponds to a 2-layer MLP trained on a noisy MNIST dataset. We demonstrate that the student underfits the hard points in our two MNIST settings, while simultaneously in Fig 20 and Fig 21 we observe that our theoretical predictions for eigenspace convergence holds true.

*by the teacher*. In this sense, the bias of distillation is an *exaggerated but non-identical* version of the bias of standard gradient descent.

In the linear setting, we find this bias to hold in all of the 15 different random pairs of eigendirections, while in the MLP setting this holds in all but two of the 15 different random pairs of eigendirections. Note that these eigendirections are picked at random from the set of all directions and not cherry picked. Specifically, the top direction is sampled at random from the top 15 directions (without replacement), and the bottom from the directions with indices in $[20, 60]$ (without replacement).

To compute the eigendirections, in the case of the random features setting, we compute the directions of the random-features-transformed data. We then project the weight matrix $\mathbf{W}$ along the eigendirection $\mathbf{v}$, and then take the $\ell_2$ norm $\|\mathbf{W}^\top \mathbf{v}\|_2$ to compute the projection. Note that in the theory, we dealt with a model with a scalar-valued output, and so $\mathbf{W}^\top \mathbf{v}$ would have been a scalar. In the case of the 2-layer MLP, we directly compute the eigendirections of the input data. We then take the *first layer* matrix $\mathbf{W}$ and compute the projection similarly.

**Takeaway.** We find that distillation leads to an exaggerated bias in terms of the rate of convergence along various eigendirections. This happens even in a setting trained with cross-entropy loss, and with a non-linear neural network, going beyond our theoretical assumptions. Thus, our insights from the linear regression setting in Sec 4.1 apply to a wider range of settings. We also find that underfitting happens in these settings, reinforcing the connection between the eigenspace regularization effect and underfitting.

**Other details.** For the random features setting, we train on a subset of 128 datapoints with 5000 ReLU random features. The training data has 0.25 probability of a mislabeling. We use a batch size of 16 and learning rate of 0.001. Both teacher and student are trained for 40 epochs, and the student with a temperature of 5.

In the MNIST setting, we use a 2-layer MLP with 1000 hidden units trained on a subset of 128 datapoints with no label noise. We use a batch size of 16 and learning rate of 0.0001. Both teacher and student are trained for 20 epochs; the distillation loss uses a temperature of 4. All other details are identical to the previous setting.

## C.5    Verifying the gradient space view in practice

We now demonstrate that the conclusions of Theorem 4.2 indeed hold good empirically (even in a settings where our theoretical assumptions don't exactly hold e.g., we will use mini-batch instead of full-batch). To do this, we build a synthetic perfectly classifiable dataset (very similar to that used in Theorem 4.2) with class similarities encoded at the logit level. We then show that distillation helps in better aligning the weights of that dataset, thus verifying that it has experienced "denoised" gradients.

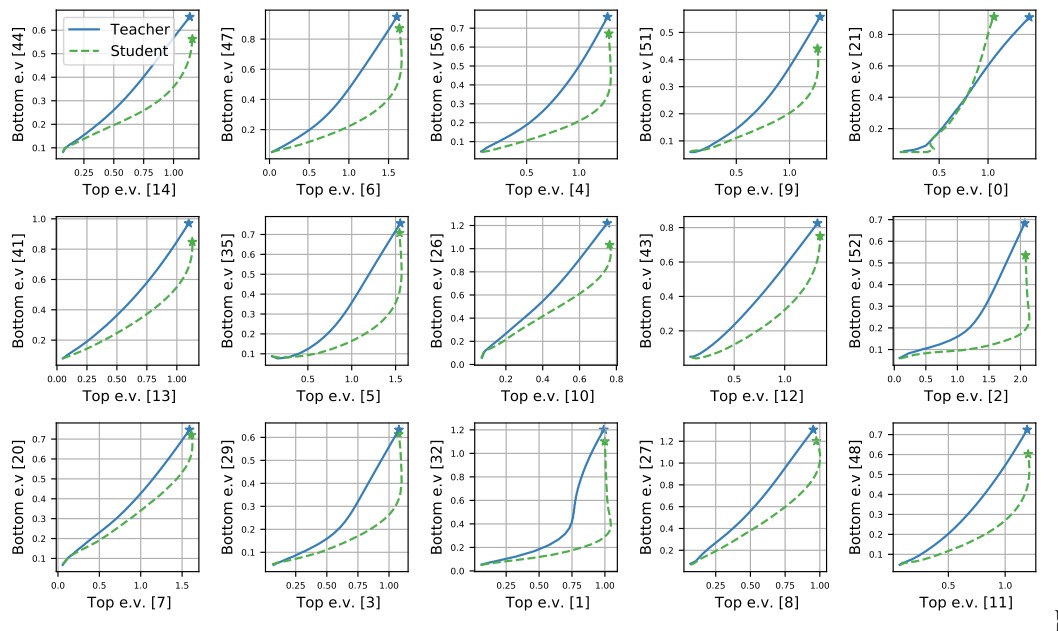

Figure 20: **Eigenspace convergence plots verifying the eigenspace theory for NoisyMNIST-RandomFeatures setting**: In all these plots, the $X$ axis corresponds to the top eigenvector and the $Y$ axis to the bottom eigenvector (see Sec C.4 for how they are randomly picked). Each plot shows the trajectory projected onto the two eigendirections with the $\star$ corresponding to the final parameters. In all cases we find that both the student and the teacher converge faster to their final $X$ value, than to their $Y$ value showing that both have a bias towards higher eigendirections. But importantly, this bias is exaggerated for the student in all cases, proving our main theoretical claim in Sec 4.1 in a more general setting with multi-class cross-entropy loss, finite learning rate etc.,

**Dataset details.** We consider a $K$-class classification dataset $(K = 10)$ where the $i$th datapoint's features can be written as a $K$-"channel" input $\mathbf{x}_i = (\mathbf{x}_1^{(i)}, \ldots, \mathbf{x}_K^{(i)})$ where each channel $\mathbf{x}_k^{(i)} \in \mathbb{R}^D$ is $D$-dimensional $(D = 100)$. For each $k$, we pick a global "ground truth" class vector $\boldsymbol{\mu}_k^\star$ sampled from the $D$-dimensional normal distribution. Assuming a uniform distribution over $K$ classes, given a label $y$, we generate $y$th channel input $\mathbf{x}_y$ as $\mathbf{x}_y = \alpha \boldsymbol{\mu}_y^\star + (1 - \alpha)\mathbf{z}_y$, where $\alpha = 0.01$ and $\mathbf{z}_y$ is uniformly sampled from the $D$-dimensional unit hypersphere truncated to $\mathbf{x}_y \cdot \boldsymbol{\mu}_y^\star > 0$. Here, the $\alpha \boldsymbol{\mu}_y^\star$ is added so as to provide sufficient non-zero margin for the points from the decision boundary.

Next, we also set the co-ordinates corresponding to a few other classes. We first randomly pick 3 other classes meant to be "similar non-target" classes. Then for each such class $k$, we set $\mathbf{x}_k = \beta(\alpha \boldsymbol{\mu}_k^\star + (1 - \alpha)\mathbf{z}_k)$ where $\alpha$ and $\mathbf{z}_k$ are sampled as before, and $\beta$ is a random value rescaled Beta distribution with parameters $a = 4$ and $b = 1$ rescaled by a multiplicative factor equal to $0.8 \cdot (\mathbf{x}_y \cdot \boldsymbol{\mu}_y^\star)$. This scaling factor ensures that $\mathbf{x}_k \cdot \boldsymbol{\mu}_y^\star$ is smaller than the ground truth margin $\mathbf{x}_y \cdot \boldsymbol{\mu}_y^\star$), and so the dataset remains linearly separable.

Besides the target class and the 3 non-target similar classes, all other co-ordinates are set to be zero for $\mathbf{x}$.

**Training details.** For training, we consider a linear classifier that has $K$ output nodes, each drawing input from the corresponding $D$ dimensions of its class. We use Adam with a learning rate of $0.1$, with batch size $128$, and $512$ training datapoints. We train both teacher and student for $10$ epochs, and the student with temperature $5$.

**Observations.** In this setting, although both the teacher and student have high accuracy, the self-distilled student outperforms the teacher by about $0.7\%$ (see Fig 23 left).

Next, we compute the number of $(\mathbf{x}, k)$ pairs in the training set whose gradients have the correct sign under either loss. For each point $\mathbf{x}$ and each node $k$, we check whether $\mathbf{x}_k \cdot \boldsymbol{\mu}_k^\star > 0$ if and only

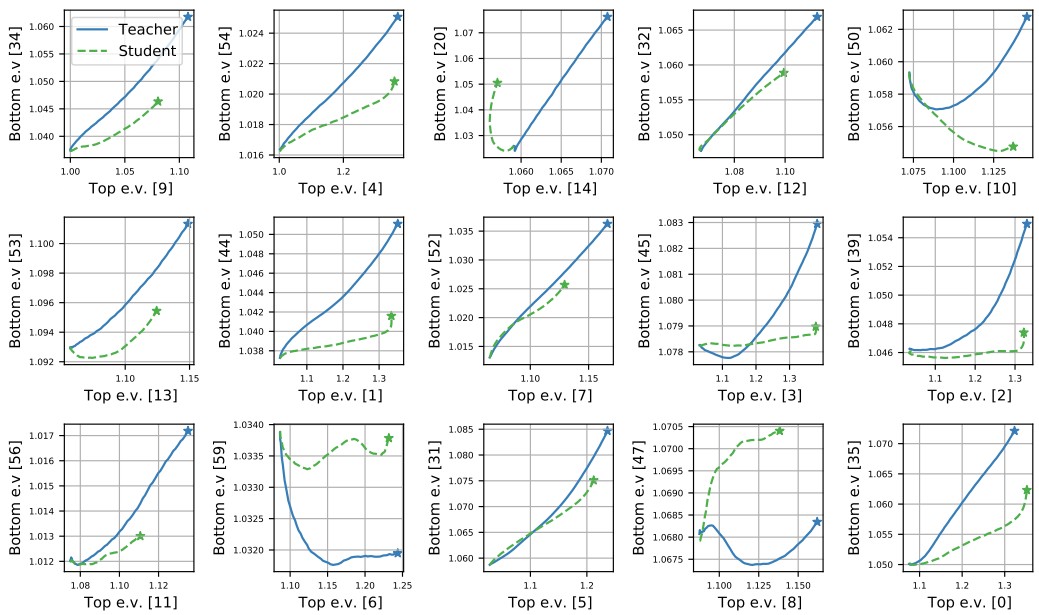

Figure 21: **Eigenspace convergence plots verifying the eigenspace theory for MNIST-MLP setting** : In all but two cases , we find that the student converges faster to the final $X$ value of the teacher than it does along the $Y$ axis. This demonstrates our main theoretical claim in Sec 4.1 in a neural network setting.

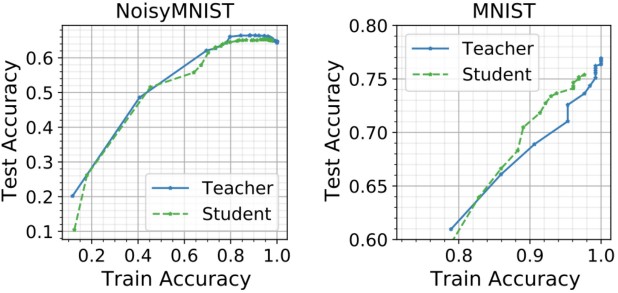

Figure 22: **Test/train accuracies for MNIST settings for Sec C.4:** In $X$ axis, we plot the training accuracy, and in the $Y$ axis the test accuracy, computed at various points of time in the trajectory. We note that in the linear case (left), there is little difference in the accuracies of the student and the teacher, likely because this is a very simple setting where ignoring the lower eigendirections (as seen in Fig 20 to distillation) has little effect. But for the MLP setting (right), we find that the student achieves higher test accuracies than the teacher for a given training accuracy. This is evidence that the student uses "better" directions (i.e., top eigendirections) to fit the data.

if the target probability under the given loss is non-zero. This can verify our key proof idea that distillation denoises the gradients even in the absence of explicit noise in the dataset. According to this computation, we find that for one-hot loss, only $73.22\%$ of such pairs have the right sign of gradient, while for distillation this is $97.77\%$, thus verifying that distillation is indeed able to denoise.

Inspired by the formulation in Theorem 4.2, we then analyze the cos-similarity between the weights learned for each class, and its corresponding class vector. We find in Fig 23 middle that the student has a better alignment than the teacher, despite the fact that the student is trained with the teacher's logits (and no extra information is given). This proves the main result of Theorem 4.2. Thus, even in a perfectly classifiable dataset, we are able to make the student outperform the teacher because of class similarities at the logit level.

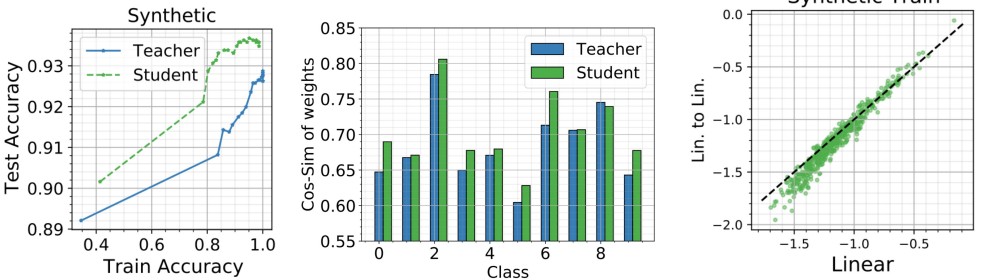

Figure 23: **Plots verifying the gradient space view in a synthetic dataset:** On the **left**, we show the trajectory of test and train accuracies for the teacher and student, demonstrating that distillation indeed helps in our non-noisy, perfectly-classifiable dataset. In the **middle**, for each class, we plot the cos-similarity of the weights with the ground truth class mean. We find that the student consistently has higher alignment with the ground truths than the teacher; this demonstrates that distillation has the ability to denoise the gradients (and outperform the teacher). In the **right**, we report logit-logit scatter plots demonstrating the hard-point-underfitting effect. Specifically here, easier points are overfit, while harder ones are more likely to be underfit.

Notably, we also observe the underfitting phenomenon in this setting (see Fig 23 right). This suggests that the underfitting phenomenon is indeed connected to how distillation denoises gradients in the presence of class similarities.

