# OpenReview forum: "On student-teacher deviations in distillation: does it pay to disobey?"
_ICLR.cc/2023/Conference — Submitted to ICLR 2023_

### Official Review · Reviewer_M824 · 2022-10-24

**Confidence:** 4
**Clarity, Quality, Novelty And Reproducibility:** 1. The paper is clearly written.
2. …
**Correctness:** 2
**Technical Novelty And Significance:** 3
**Empirical Novelty And Significance:** 2
**Recommendation:** 6

**Details Of Ethics Concerns:**

No ethics concerns.

**Strength And Weaknesses:**

Strength:

1. The authors make connections between the theoretical understanding and empirical observations in distillation, and provide an deeper and unified perspective of regularization and denoising.
2. The empirical observations are extensively conducted on various datasets and models.

Weakness:

1. Explaining under-fitting via eigenspace sparsification is insufficient. The empirical observations are based on non-linear models while the theorem holds for linear models. The connection between the hard samples and lower eigendirections is also unclear.
2. The theorem for the gradient denoising view is loosely connected with the loss-switching experiments. It might be more concrete to show one-hot update suppresses the similar but non-target logits, and how that produces destructive gradients and hurt generalization.
3. The proposed idea provides some insights but is not totally novel. It mainly generalizes existing views in wider settings, e.g., GD-optimization case and clean label case.


**Summary Of The Paper:**

This paper first characterize the deviations between the teacher and the student; then provide theoretical perspectives on how distillation induces such deviations. Specifically, they first show that (1) the student under-fits on hard samples and (2) a large deviation in the early stage can be recovered by distillation introduced in the later stage. Then they provide an explanation for (1) by viewing distillation as an eigenspace sparsifier in GD-based linear model and an understanding for (2) by viewing distillation as a gradient denoiser.

**Summary Of The Review:**

This paper provides an deeper and unified view of eigenspace sparsification and gradient denoising for distillation. The empirical results are extensive. However, the theoretical results are limited to linear models; Some claims are lack of evidence and are loosely connected with the empirical results (See weakness).

---

> ### Author Response · Authors · 2022-11-17
> **New experiments verifying our insights in practice & why our work makes significant advances over existing theory**
>
> Thank you for your time in reviewing our work. We're happy to see that you view our work as providing a deeper & unified perspective of distillation.
>
> We understand you have concerns about the relationship between our theory and our empirical settings. We’ve added concrete experiments to alleviate these concerns completely:
>
> > Explaining under-fitting via eigenspace sparsification is insufficient. The empirical observations are based on non-linear models while the theorem holds for linear models. The connection between the hard samples and lower eigendirections is also unclear.
>
>  **We have added an MNIST + MLP experiment in Sec C.4 reporting that the underfitting phenomenon occurs here together with the eigenspace regularization.** This clearly demonstrates that our insight regarding eigenspace regularization does translate to the neural network (NN) regime. Thank you for the useful feedback which led us to add this valuable experiment!
>
> Next, we'd like to clarify that the connection between lower eigenvectors and noisy/hard examples is well-known and considered intuitive e.g., Li et al., 2020; Dong et al., 2019; Arpit et al., 2017 can shed light on this further.
>
> We must also emphasize that while we don't prove formal results about NNs, **our theory goes significantly beyond what existing theory has done in bridging gaps to practice. (please see end of this response)**
>
> > theorem for gradient denoising view is loosely connected with the loss-switching experiments... more concrete to show one-hot update suppresses the similar but non-target logits, and how that produces destructive gradients and hurt generalization.
>
> - Addressing this, **we've added experiments in Fig 16** showing how persistent one-hot training for a long time does result in _destroying_  distillation's gains (despite weight regularization), while persistent distillation does not.
> - We've also **added experiments in Sec C.5**. This is an empirical version of Thm 4.2 setting, demonstrating the effect of destructive gradients.
>
> > ... provides some insights but not totally novel... mainly generalizes existing views in wider settings, e.g., GD-optimization case and clean label case.
>
> **We'd like to strongly push back on this criticism: we require significantly different technical arguments & identify important novel insights not present in previous work as we explain below.** In brief:
> - The underfitting of probabilities and how it relates to eigenspace regularization is novel.
> - Our eigenspace view result makes a connection to early-stopping in the context of GD that is completely opposite to the connection made in Mobahi et al.,
> - The proof technique for the denoising effect in the absence of noise involves GD and has nothing to do with the much simpler non-GD arguments in Menon et al., line of work.
> - We’ve also taken significant steps towards connecting theory to practice which prior work has not done.
>
> In more detail:
> - **The connection of distillation with underfitting of probabilities is completely novel & does not exist in prior work.** While Mobahi et al., identify eigenspace sparsification in a non-GD setting, they don't empirically connect it to underfitting, or verify the sparsification effect in practice.
> - **We provide a novel insight _clarifying_ the effect of early-stopping**: While Mobahi claim that early-stopping & KD have opposing effects in their non-GD setting, we show that in contrast, these have similar effects: KD _amplifies_ the same effect of early-stopping! This clarification is important since early-stopping is highly relevant to distillation practice  [(Dong et al., 2019; Cho & Hariharan, 2019; Ji & Zhu, 2020, Wang et al., https://arxiv.org/abs/2210.06458 ].
> - The proofs in Menon et al., are agnostic to GD as they look at a bias/variance decomposition of an abstract function. **Our proof involves a completely different technical argument** that looks at the dynamics of gradient descent. This is what allows us to remove the label noise assumption, which was impossible with the Menon et al., setting.
> - We have not just generalized these insights but **unified the two lines of work (please see Sec 4.3)** by showing how the two views are related to each other.
> - We have also taken significant efforts efforts towards connecting these insights to empirical settings in ways prior work has not done:
>     - **The eigenspace regularization was not demonstrated in practice in prior work which we now demonstrate in an added experiment in an MLP model.**
>     -  to the best of our knowledge, prior work has not examined as diverse **a variety of experimental settings as ours (>6 different architectures and > 8 different image, language datasets)**.
>
>
> We strongly hope that our added experiments relating theory to practice, and our argument showing why our results generalize prior work **in a novel and non-trivial way** — and not just merely extend upon them — will encourage you to reconsider your score for our paper!

---

> > ### Comment · Reviewer_M824 · 2022-12-06
> > **Thanks for your responses**
> >
> > Thanks for providing the new experiment results and clarifying the novelties. The responses addressed my concerns and I would like to raise my score to 6.

---

> > > ### Author Response · Authors · 2022-12-07
> > > **Thank you!**
> > >
> > > Dear reviewer,
> > >
> > > Thank you for acknowledging that our new experiments and clarifications address your concerns. Also, thank you for raising your score!

---

### Official Review · Reviewer_njgH · 2022-10-24

**Confidence:** 4
**Correctness:** 3
**Technical Novelty And Significance:** 3
**Empirical Novelty And Significance:** 4
**Recommendation:** 8

**Clarity, Quality, Novelty And Reproducibility:**

The paper is generally well written. Some figures can be improved with better labels and auxiliary information to help the reader. In terms of novelty, the paper presents an interesting instance-level analysis of teacher v/s student logits as well as theoretical arguments to explain the intuition behind these observations. The authors also share the hyperparameters used to reproduce the results, as well as detailed proofs for the theorems.

**Strength And Weaknesses:**

**Strengths**
1. The paper provides an interesting and novel approach to analyze the deviations between teacher and student logits. While others (Mohabi et al., Yuan et al.) have shown regularization effects due to distillation, the experiments here help us understand the form that the regularization takes in terms of the predicted logits.
2. The results on switching between KD and standard training also provide usable insights into when KD gradients help and may result in better KD scheduling and training algorithms.
3. The experiments analyse a variety of settings for both KD, and self distillation.
4. The discussion of the related work is comprehensive and the paper clearly situates itself among existing literature.

**Weaknesses/Questions**

1. It might be useful to include the test and training accuracies (or differentiate between correct/incorrect examples) in all the figures.
2. Does the underfitting depend on how long the student is trained? I am curious to see if the same holds if the student is trained longer than the teacher.
3. Fig. 12 actually shows an interesting insight in qualifying which points the two classifiers get wrong. In fact, given that the "harder" points tend to be generally misclassified, the argument that the underfitting leads to better generalization may be flawed. If the authors could provide some numerical values for the fraction of underfit points in every case, we might be able to differentiate between the effect of distillation v/s that of optimization error. Thm. B.3 also appears to point to early stopping of the student being an important criterion.


**Summary Of The Paper:**

The paper studies knowledge distillation by investigating the points where the student deviates from the teacher's predictions. The authors suggest that the success of knowledge distillation is because student networks underfit ``hard'' points. The authors provide empirical evidence by comparing student logits with that of the teacher for a variety of settings. They also show a very interesting phenomenon -- switching losses from standard cross-entropy loss to a distillation based loss recovers the same properties that a model trained with distillation alone. The paper also provides theoretical intuition for this by extending XTK style arguments to show that KD acts as a regularizer. Additionally, the authors also provide a gradient space view suggesting that in the linear setting, distillation loss gradients "denoises" the effect of negative examples with similar features.

**Summary Of The Review:**

The paper explains KD as a regularizer that helps student networks underfit "harder" points and therefore make different mistakes from the teacher. The experiments are insightful and reveal previously unstudied behavior in student networks. While there may still be some confounding factors due to early stopping, and certain inconsistencies for language models, the paper still adds value in terms of understanding where distillation may be improved. The authors have conducted in-depth experiments to support their arguments.

---

> ### Author Response · Authors · 2022-11-17
> **Thanks for the detailed and positive review!**
>
> Thank you for your enthusiastic review of the paper. We are happy that you’ve carefully understood and appreciated the novelty and significance of our results.
>
> > It might be useful to include the test and training accuracies (or differentiate between correct/incorrect examples) in all the figures.
>
> We had listed the test accuracy of all the plots in Table 2. We have now added the training accuracy.  For the sake of reduced clutter, we have currently refrained from adding them directly to the plots.
>
> > Does the underfitting depend on how long the student is trained? I am curious to see if the same holds if the student is trained longer than the teacher.
>
> This is a fantastic question. **We added an experiment where we train 2x long and found that underfitting persists.** Please see C.3 for further discussion and Fig 17 left two images.
>
> > Fig. 12 actually shows an interesting insight in qualifying which points the two classifiers get wrong. In fact, given that the "harder" points tend to be generally misclassified, the argument that the underfitting leads to better generalization may be flawed.
>
> This is a great question, and we are pleased that you’ve also evaluated the appendix in detail. **We however believe that this does not contradict “underfitting ⇒ better generalization”**. We believe it’s possible for the underfitting to have “given up” on certain harder/noisier examples (and misclassify them) only to gain in accuracy on a different larger fraction of examples (overall, improving generalization). There are multiple ways to intuit about this:
> - The eigenspace regularization from Thm 4.1 implies that KD deprioritizes tail directions. While this helps us rely on simpler and more accurate directions, it may also forget some tail points that require harder features.
> - Thm 4.2 tells us how distillation can “source” gradients for a node k from a point $(x,y)$ that has soft similarities to class k; furthermore, this may help the student correctly classify a point $(x', k)$ belonging to class k. In this situation, the student would underfit the teacher’s prediction on both points, but in one case it would reduce the target class probability and in the other increase it. The misclassification of the former is crucial to better classify the latter.
> - An alternative possibility is that the ground truth labels are themselves noisy to some extent e.g., recent work [https://arxiv.org/abs/2006.07159] has found that imagenet dataset may have incorrect labels. Thus, some of the labels the student achieves may indeed be “correct” labels, although recorded as misclassified.
> We summarize this discussion in C.2.2.
>
> > If the authors could provide some numerical values for the fraction of underfit points in every case, we might be able to differentiate between the effect of distillation v/s that of optimization error.
>
> Thanks for this valuable suggestion. In the current paper, we have focused on _qualitatively_ establishing this phenomenon across a variety of settings. We observed that the hard-point-underfitting manifests in a variety of different ways across different datasets, which makes it tricky to compute a single quantity that faithfully captures the phenomenon across the board.
>
> We must note that we had introduced the noisy dataset setting as a controlled setup where it is easier to understand the relationship between underfitting and improved generalization: in particular, here we observed that the student underfits all of teacher’s mislabeled points, thereby denoising the dataset.
>
> But we agree that having a quantity might be ideal so we will try to devise a careful quantification for the final version of the paper.
>
>
> > “Thm. B.3x also appears to point to early stopping of the student being an important criterion” and “While there may still be some confounding factors due to early stopping”
>
> - We’d like to clarify that in our view, early-stopping _is_ indeed an important factor enabling distillation to work.
> - Our point is that while early-stopping already has its own bias, distillation amplifies this effect. This causes distillation to traverse parts of the parameter space that early-stopped GD alone would not. Please see Fig 20/21 for a visualization!
> - Early-stopping has time and again appeared in various works on distillation [Dong et al., 2019; Cho & Hariharan, 2019; Ji & Zhu, 2020, Wang et al., 22], so we believe our incorporation of early-stopping in our theory is not only warranted, but helps formalize existing empirical intuition.
> -  It is also worth noting that Mobahi et al., suggest that early-stopping and distillation provide opposite effects, which is an artifact of their non-GD setting. However, because we study GD, our work is able to argue that the effects are similar yet subtle different (e.g., KD exaggerates the effect of early-stopping).
>
> We hope we have addressed all your concerns! Thanks again for the positive review.

---

> > ### Comment · Reviewer_njgH · 2022-12-04
> > **Response to Authors**
> >
> > I thank the authors for their detailed responses to all the concerns. While I agree with reviewers xDEX and M824 about the some of their concerns, the authors have clearly pointed out the differences from existing work. In addition, the experimental evaluations (especially the updated ones) address most of my concerns.
> >
> > I am therefore keeping the score as it is. However, I do invite other reviewers to engage with the paper, as it appears that the authors have made significant changes to address their concerns.

---

> > > ### Author Response · Authors · 2022-12-05
> > > **Thank you for acknowledging our detailed responses to all concerns**
> > >
> > > Dear reviewer, thank you for going over the other reviews, our detailed responses and acknowledging our efforts to clarify all the concerns. We appreciate it! We look forward to the updates from the other reviewers.

---

### Official Review · Reviewer_EHbp · 2022-10-25

**Confidence:** 3
**Correctness:** 3
**Technical Novelty And Significance:** 2
**Empirical Novelty And Significance:** 3
**Recommendation:** 6

**Clarity, Quality, Novelty And Reproducibility:**

Clarity, Quality: Good. The paper is well-written with clear explanations about each part.

Novelty: Good. The view of the paper is novel while the methods used for analysis are not.

Reproducibility: Fair. The supplementary code is not provided.


**Strength And Weaknesses:**

**Strengths**
+ The paper is well-written and easy to follow.
+ The observations are drawn from comprehensive experiments with multiple architectures and datasets.
+ The experiments on and results of loss switching are interesting.


**Weaknesses**

- It would be more helpful if the author could conclude some take-home messages from their work and give a clear answer to the question in the title. It will help readers to understand the paper better and inspire further research on the topic of knowledge distillation.

- As shown in Figure 4, it looks like KD from the intermediate steps helps train the student better. Is it possible to get some practical suggestions from these experiments?

- This paper works on the distillation of the predicted logits/probabilities, however, the feature distillation has drawn more attention recently. To some degree, it should also affect the prediction of the logits/probabilities. It would be more interesting to include some analysis of the feature distillation methods.

**Summary Of The Paper:**

In this paper, the authors first observe that in most cases, the student underfits points that the teacher finds hard by experiments across image and language classification with self- and cross-network-distillation settings. Then they analyze how the loss switching affects. Afterward, the student-teacher deviations are formalized in both eigenspace and gradient-space views. They further are used to help understand the knowledge distillation methods from different views. Overall, the contributions of this paper bridge the theory and practice in the field of knowledge distillation.


**Summary Of The Review:**

The paper shows comprehensive experimental and theoretical analysis. It also sheds light on potential research in the future on knowledge distillation. It would be better to improve the manuscript according to the above weaknesses.

---

> ### Author Response · Authors · 2022-11-17
> **Thanks! Key takeaways and practical ideas**
>
> Thank you for taking the time to review our work. You have raised some interesting and important big picture questions, which we’ve addressed below.
>
> > It would be more helpful if the author could conclude some take-home messages from their work and give a clear answer to the question in the title. It will help readers to understand the paper better and inspire further research on the topic of knowledge distillation.
>
> Thanks for the suggestion. We have updated the conclusion and the introduction to highlight our takehome messages.
>
> Concretely:
> - **We argue that student-teacher deviations in probabilities can be good**, which has important implications for practitioners trying to bridge the teacher-student gap, an important area of distillation research.
> - Understanding of distillation is still lacking and our theory bridges key gaps with practice, showing how distillation behaves **under gradient descent. In particular, it induces a regularization effect and a denoising effect.**
> - We also formalize and **emphasize the importance of early-stopping in distillation**, which has been given significant attention in empirical work on distillation [(Dong et al., 2019; Cho & Hariharan, 2019; Ji & Zhu, 2020, Wang et al., 2022].
>
> > As shown in Figure 4, it looks like KD from the intermediate steps helps train the student better. Is it possible to get some practical suggestions from these experiments?
>
> This is an exciting question. Indeed, we expect our work to inspire better approaches, which is something we are working on as well. Specifically, there are also many important follow-ups one could think of from each of our sections/insights:
> - **Based on Sec 3.2:** As you suggest, one could explore **different schedules**: our experiments overall suggest beginning with one-hot, then following with KD (and perhaps perform one-hot towards the end for a short while).
> - **Based on Sec 3.1:** Our underfitting observations suggest that we may want to introduce **importance weighted distillation** that allows the student to deviate on harder points.
> - **Based on Sec 4.2’s insights** on destructive gradients in one-hot training: can we smooth one-hot training in a clever way so as to **reduce destructive gradients** on similar non-target classes? Is there a way to relax one-hot optimization?
> - **Based on Sec 4.1 analysis:** Can we analyze other distillation techniques using our early-stopping framework? Can we develop distillation techniques that **further amplify the reliance on the top eigenvectors**?
>
> **All these are open questions we are investigating, which are beyond the scope of the current work.** In this paper, we wished to focus on the fundamental questions underlying “understanding distillation” **which is already poorly understood especially in the context of GD**, and to which we make multiple key contributions.
>
> > This paper works on the distillation of the predicted logits/probabilities, however, the feature distillation has drawn more attention recently. To some degree, it should also affect the prediction of the logits/probabilities. It would be more interesting to include some analysis of the feature distillation methods.
>
> This is a very interesting follow-up question. We believe that this may intuitively result in a “per-layer effect” where each layer’s weight matrix may undergo its own eigenspace regularization due to its corresponding KD loss.
>
> However, we must emphasize that the community’s understanding of standard distillation itself is already lacking. **We believe that the diversity of settings we have explored here (>6 different architectures and > 8 different image & language datasets) and the multitude of insights we have is already a significant first step towards bridging theory and practice.** We hope to explore other variants of distillation in future works which we agree is an important direction.
>
> Overall, we hope that
> - our clarification of the takeaways from the work,
> - the many future directions that it can inspire (see also Rev 3, 2nd point)
> - the depth and breadth the paper has already covered, both theoretical and empirically
>
> help you rate the paper more positively towards acceptance. Thanks once again for engaging with the paper!

---

### Official Review · Reviewer_xDEX · 2022-10-25

**Confidence:** 4
**Correctness:** 2
**Technical Novelty And Significance:** 2
**Empirical Novelty And Significance:** 1
**Recommendation:** 5

**Clarity, Quality, Novelty And Reproducibility:**

Paper needs to be re-written to highlight the important and coherent empirical observations rather than throw all the plots at the reader in the hopes that they would draw their own conclusions. Since this work analyzes empirical behavior, it would greatly improve the quality of the work if it runs through a single illustrative example and show-case various observations one would see in almost all the settings.

Suggestion for improving the paper writing / clarity?
 - Instead of showing all the plots at once (abundance of information)
 - Just pick one setting (say two architecture combinations Res56-56 and Res56-20, one instance of self-distillation and cross-distillation)
 - Go through all the observations at once
 - This would make the paper much more readable and clearly highlight the important observations
 - You can put the remaining plots in the experiments section to show that above results are applicable in other setups
 - Currently, things are just all over the place


Nit-Pick:
- Page 3, Eq.1, $\ell(f(x_n))$ does not incorporate a label $y_n$. It is unclear how this is a loss function?
- Page 9, Para.1, " .. teacher's probabilities Menon et al. (2021). ... "


**Strength And Weaknesses:**

Strengths:
- Problem being studied is of importance to the distillation community.
- Empirical observations and lots of analytical experiments could help understand the distillation mysteries.

Weaknesses:
- Theoretical analysis seems to be done on very narrow setup, for ex.,
   - Thm4.2 studies a linear classifier on top of a specially constructed K-class problem. It is unclear how their observations extend beyond this setup. Even in loss switching there are many instances where switching to one-hot loss from KD does not seem to hurt the performance. So their point about One-Hot getting stuck at the optimal solution does not seem to apply here.
   - Thm4.1 talks about KD enabling the student to learn the "nice" directions first. It is not clear why this is something special to the KD. Some recent works have shown that SGD in general learns functions of increasing complexities (see https://arxiv.org/pdf/1905.11604.pdf )


- Paper writing makes it very hard to digest the crux of this work. There does not seem to be any coherency in the empirical observations being presented.




Questions for Authors:

(1) How is the teacher network trained in the self-distillation setup?

(2) In Fig.1 and 2,
 - How do you define hard points? What is the measure of hardness used to classify a data point being easy or hard for a model? In CIFAR-100 Resnet56-56, it seems half the points lie above the Y=X line and half of them lie below this line.
 - What are the accuracies of the teacher and student model?
 - For the train data, do you use data augmentations? If so, do you see similar behavior on the augmented data?
 - In which of these instances, the teacher or the student fit the train data without any error?

(3) I don't understand the conclusions from NoisyCIFAR100 experiments.
 - Teacher is trained with noisy labels
 - Student behaves similar whether the teacher had clean or noisy labels (Fig1a -- first and last plot, Fig2 -- second and last plot)
 - Student just underfits the teacher softened probabilities -- this has been the case so far in all the cases
 - In the noisy case, the teacher had more scores (X values) that are small -- simply due to the fact that the teacher was presented noisy labels in training
 - How does this prove that the "underfitting is good"? or that "smaller student models go beyond a larger teacher"?

(4) Why is the monotonic logit transformation used $\log \frac{u}{1-u}$ ? Do these plot hold for other transformations? Say linear, or other metrics like KL-divergence between teacher and student probabilities?

(5) What is the definition of "Fidelity" in Sec.3.1?

(6) Could the authors summarize what is going on the Sec.3.2?
 - The setup tries to switch from One-Hot training to Distillation or the vice-versa in the middle of the training. What was the aim of this experiment? Figure 4 shows that if you switch from One-Hot to Distillation you are better off doing so as early as possible? Is that not just saying that learning from softened probabilities is better than using one-hot labels? Please elaborate if I missed some point.

 - Btw, I would exclude Fig4(3) from conclusions purpose simply due to the fact that the accuracies are all close by for almost all distillation experiments ( 0.663 - 0.667 ) which might just be due to the variance in the experimental runs.

(7) Thm.4.1 -- applies to linearized NTK regime for neural networks, which would be applicable for neural networks with width going to infinity. It is unclear how you would apply it to the neural networks setup. Besides, what are nice directions? It is unclear how Thm.4.1 explains anything?  Some recent works have shown that SGD in general learns functions of increasing complexities (see https://arxiv.org/pdf/1905.11604.pdf )

(8) Concrete Example in Thm.4.2
 - Do you have experimental results on this example? Since the theorem 4.2 states that you get better quality updates from distillation does this mean CE is unable to recover the correct classifier? Are there any quantitative / qualitative results for this example?

 - Thm4.2 studies a linear classifier on top of a specially constructed K-class problem. It is unclear how their observations extend beyond this setup. Even in loss switching there are many instances where switching to one-hot loss from KD does not seem to hurt the performance. So their point about One-Hot getting stuck at the optimal solution does not seem to apply here.

(9) How are the hyper-parameters tuned? In appendix Table.1 shows that CIFAR dataset uses peak learning rate of 1.0. Is this number a typo? This seems to high for the CIFAR datasets. Besides, even the batch size (1024) seems sub-optimal. Typical setup would include batch size (128-256) and learning rate of 0.1 for SGD+Momentum optimizer.


**Summary Of The Paper:**


This work characterizes the nature of deviations between teacher and student network during knowledge distillation (self as well as cross architectures).

In the first part, this work studies these properties empirically and notes that the following two observations:
 - Student underfits points which teacher finds hard
 - Initial training phase is not crucial for distillation benefits. One can switch from 0-1 labels to teacher's soft predictions somewhere in the middle or later parts of the training and achieve similar results as training with teacher predictions from the beginning.

In the second part, it develops following two theoretical viewpoints to explain this observed behavior:
- Distillation as regularizer in the eigenspace: It analyzes a continuous-flow gradient descent model on linear regression with early-stopping. Thm4.1 says that student relies less on the bottom eigendirections than the teacher under the condition that both models have converged well along the top eigendirections.


- Distillation as denoiser of gradients : It constructs a K-class classification problem with each input being K channel and each channel drawn from a multivariate Gaussian with co-variance being a scaled identity matrix. It considers a linear architecture on top of this construction. Thm.4.2 shows that distillation's gradients are more optimal than one-hot gradients and also more optimal than the teacher's weights themselves.



**Summary Of The Review:**

This work characterizes the nature of deviations between teacher and student network during knowledge distillation (self as well as cross architectures). Although the problem at hand is interesting, its execution from both empirical and theoretical standpoint seems below par. In addition, their seems to be incoherency in the paper writing that obfuscates main observations and their connections with the thoery developed in the paper.


-----

I've updated my score post the rebuttal.

---

> ### Author Response · Authors · 2022-11-17
> **Part 3: Answering your other clarification questions**
>
> > (2) I don't understand the conclusions from NoisyCIFAR100 experiments... How does this prove that the "underfitting is good"? or that "smaller student models go beyond a larger teacher"?
>
>
> - Yes, you're right, underfitting happens in both the noisy and clean data setting.
> - In the noisy dataset, we argue that **the underfitting corresponds to denoising**: all the mislabeled data lie below the Y=X line, meaning that the student underfits the teacher’s probability of the _incorrect_ class. This “underfitting = denoising” is what suggests that underfitting must be good. Furthermore in Sec 4.2, we argue how distillation denoises even when there’s no explicit label noise thus making this intuition more formal and general! This also demonstrates how the underfitting phenomenon and the gradient denoising viewpoints are closely related, which we believe adds to the coherence of our results. We’ve rewritten this in the paper.
> - Regarding “smaller student models go beyond larger teacher”, small models typically achieve less test accuracy than the teacher (see Table 2). Yet, Fig 2 suggests that smaller students do denoise the teacher’s wrong labels. This implies the smaller student fits better than the teacher at least on the small (noisy) subset of the data, even if not in a global average sense.  Note that **this intuition is absent in distillation literature** as the belief is that "small students underperform w.r.t. the teacher" which we argue is too reductive.
>
> > (4) Why is the monotonic logit transformation used log⁡ u/1−u? Do these plot hold for other transformations? Say linear, or other metrics like KL-divergence between teacher and student probabilities?
>
> This is similar to plotting a log-log plot, which we do for the sake of visual clarity. **We have now added Fig 18 where we plot a bunch of other metrics** of the kind you have suggested (valuable suggestion, thanks!). We find that even KL divergence shows similar “hard points are underfit” behavior!
>
> > (5) What is the definition of "Fidelity" in Sec.3.1?
>
> We used  “fidelity” broadly to refer to “how well the student captures the teacher probability”. We have replaced this with "deviations" to avoid confusion.
>
> > Btw, I would exclude Fig4(3) from conclusions purpose simply due to the fact that the accuracies are all close by for almost all distillation experiments ( 0.663 - 0.667 ) which might just be due to the variance in the experimental runs.
>
> Thanks for the very careful observation! However, we had plotted the 95% confidence interval (based on Student’s t distribution) from 5 runs. Even in Fig4(3) when we do KD for the first 1/4th of training, the performance of distillation is well above the standard model! So this is still strong evidence for our claim!
>
> > (9) Table 1 shows that CIFAR dataset uses peak learning rate of 1.0... seems too high
>
> First, **we’ve added an experiment in Sec C.3 Fig 17 right**, where we find that even for a batch size of 128, the same underfitting occurs.
>
> But we’d like to clarify that the use of a larger batch size in CIFAR100 has precedent in the literature. In particular, since learning rate typically scales linearly with batch size [1], and we are using a batch size that is 4 or 8 times larger than 128 or 256, it is expected that the typical learning rate of 0.1 would be increased. Similar setting can be bound here: https://docs.mosaicml.com/en/latest/model_cards/cifar_resnet.html
>
> Finally, since we are interested in “understanding distillation” rather than “improving distillation”, we did not focus our efforts on hyperparameter tuning.
>
> [1]: Goyal et al., “Accurate, Large Minibatch SGD: Training ImageNet in 1 Hour“ https://arxiv.org/abs/1706.02677
>
>
>
> > Paper needs to be re-written to highlight...
>
> Thank you for the extremely thoughtful comments. One of the reasons we presented a variety of plots in Fig 1 is that we wanted to avoid cherry-picking and make sure the reader is aware of the various ways in which the underfitting phenomenon may manifest. However, we see your point as well and will do our best to carefully incorporate your comments and rewrite these parts of the paper for final versions.  Thanks again for the actionable feedback.
>
> > Page 3, Eq.1, ℓ(f(xn)) does not incorporate a label yn It is unclear how this is a loss function?
>
> Please note that ℓ(f(xn)) is a loss vector, as mentioned in the text following Equation 1. The elements of the vector comprise the loss function ℓ(y, f(xn)) evaluated for all possible labels y. Equation 1 then selects only the ynth component of this vector.
> We use this notation to highlight the distinction to distillation in Equation 2, where we indeed consider all possible labels from this vector, but apply weights given by the teacher probabilities.

---

> ### Author Response · Authors · 2022-11-17
> **Part 2: Some more requested experiments, and clarifying critical misunderstandings**
>
>
> > Thm4.1 talks about KD enabling the student to learn the "nice" directions first. It is not clear why this is something special to the KD. Some recent works have already shown...  [and point 7]   It is unclear how you would apply it to the neural networks setup...
>
> We're afraid there's **a critical misunderstanding** here: while you’re certainly correct that SGD already learns nicer directions first as demonstrated by our Eq (3) – which is something we also verbally state below Eq (3) for the teacher (we have made it clearer) – the key effect here is that **this bias is amplified for the student**. More concretely, **we argue that KD covers a very different (better) part of the parameter space than simple GD**. We have visualized this through "convergence plots" in Fig 20 (linear model) and Fig 21 (MLP).
>  While GD already has a slower rate of convergence along bottom directions, the trajectory of KD amplifies this effect and traverses through different parts of the parameter space with higher reliance of top eigendirections.
> This is an important yet subtle insight that we have now spelled out more clearly after Thm 4.1. We are grateful that your feedback helped us add this clarifying intuition.
>
> Note that we’ve additionally cited the paper you referred.
>
>
> > Could the authors summarize what is going on the Sec.3.2? What was the aim of this experiment? ... Even in loss switching there are many instances where switching to one-hot loss from KD does not seem to hurt the performance.... Thm4.2 studies a linear classifier on top of a specially constructed K-class problem. It is unclear how their observations extend beyond this setup...
>
> **Motivation:** These experiments tell us _which_ student-teacher deviations are critical for the success distillation. We had outlined this right before and at the end of Sec 3.2, but we’ve rearranged and rewritten this now for greater clarity.
> - In particular, we rule out deviations that had been proposed in some existing work. These works argue that in the non-convex deep learning setting, there are critical non-linear student-teacher deviations in the initial phase of training — which may determine what basin/representation the training may converge to (Allen-Zhu ‘20 and Li, Phuong and Lampert ‘19, Jha et al ‘20) – and this is crucial for distillation to succeed.
> - Our main argument is this: in deep network training, **we can replace the first quarter of distillation training with one-hot, with no drop whatsoever in distillation’s gains, thus showing that these early-phase hypotheses/factors are not critical to explain the effects of distillation.**
> - We wish to add that when we train for **a sufficiently long time**, switching to one-hot does deteriorate the gains made by distillation, thus verifying the inherent destructive nature of one-hot gradients proposed by Thm 4.2 in more general settings. Please see new Fig 16.
>
>
>  > (8) Concrete Example in Thm.4.2 Do you have experimental results on this example?
>
> Great suggestion. **We have added an empirical/quantitative result (Sec C.5 and Fig 22)** showing that the prediction of Thm 4.2 is indeed true in a similar setting!
>
> > (1)  How is the teacher network trained in the self-distillation setup? How do you define hard points? similar behavior on augmented data?...
>
> - In self-distillation, the teacher is trained like in all other settings: using one-hot labels and cross-entropy loss.
> - Hard points are points where the teacher's top class probability is very low i.e., X value is very low. If a student underfits a point it means that Y < X (explained in Fig 1 caption and second para of page 4).
> - We've listed the accuracies in Table 2.
> - Regarding augmentation, good question: we do use augmentations. We find that the underfitting phenomenon replicates even in test data (see bottom row of Fig 5/6/7 etc.,), thus it is reasonable to say this should happen even in augmented versions of training data.

---

> ### Author Response · Authors · 2022-11-17
> **Part 1: Important new experiments added to address your major (and minor) concerns**
>
> Dear reviewer, Thank you for your detailed feedback. Here's a summary of how we’ve addressed your major concerns:
> - **We’ve added a neural network (Sec C.4) example** where both eigenspace regularization and underfitting occur, establishing that our theoretical insights are still relevant to the deep learning setting.
> - **We’ve added a quantitative experiment** (Sec C.5) paralleling Thm 4.2 showing how distillation denoises even in the absence of label noise. We also find underfitting in this setting, thus connecting our eigenspace and gradient space view.
> - **We've added Fig 16** showing that **_persistent_** **one-hot training undoes the accuracy gains of distillation** thus clarifying that our theory about the destructive nature of one-hot gradients does hold.
>
> **We also clarify key misunderstandings:**
> - The **“reliance of nice directions” by KD is *not the same* as that of the well-known bias of GD**.
> - The loss-switching experiments help us understand **_which_ student-teacher deviations are important for the success of distillation** – it tells us that early-phase deviations in non-convex optimization settings are not as important as previously thought.
> - We emphasize how **our theory goes beyond existing theory in being closer to practice**, in at least 3 significant ways.
>
> > ... very hard to digest the crux of this work.
>
> The crux is: While distillation is advocated as a process where the student mimics the teacher, we understand what deviations exist between teacher and student training and when/why they are good:
>
> - **what deviations exist?** Sec 3.1 identifies a novel type of student-teacher deviation: the underfitting of hard points.
> - **which deviations are relevant?** Sec 3.2 notes that there are other deviations prior work have noted, but demonstrates that these other deviations are *not* critical to the success of distillation.
> - **how do these deviations arise?** Sec 4.1 and 4.2 both provide a formalization of how optimization bias of GD causes deviations.
> - **why are these deviations good?** Both the regularization and denoising effects in Sec 4.1 and 4.2 provide complementary insights into this question. **Sec 4.3 provides a coherent picture of how these two insights (eigenspace regularization and gradient denoising) relate to each other**.
>
> Overall, we make multiple contributions that significantly **bring theoretical understanding (which is currently lacking) closer to practice**. Hopefully, our summary above provides greater coherence and helps the reviewer better appreciate our contributions. We’ve edited our contributions for improved clarity.
>
> > Theoretical analysis... on very narrow setup
>
> Agreed that our theoretical analyses are in the linear setting. **We have now added experiments in an MNIST/MLP setup  (Sec C.4) where we empirically demonstrate (a) the eigenspace regularization phenomenon together with (b) the underfitting.** We're confident that this will go a long way towards emphasizing the relevance of our insights to neural networks, so we're grateful for your feedback.
>
> Having said that, we also believe that the current evaluation fails to take into account the fact that:
> (a) prior works suffer from similar/worse limitations, and (b) we make multiple highly non-trivial advances on top of them yielding new insights:
> - We **formalize the biases of GD** while both lines of work that we unify (Mobahi et al., ‘20 etc., and Menon et al., 21 etc.,) do not say anything about GD. Implicit biases of GD are highly relevant in the modern overparameterized era.
> - We **remove artificial noise assumptions** from [Menon et al., 2021; Dao et al., 2021; Ren et al., 2022; Zhou et al., 2021] using a completely different proof technique. As a novelty, this reveals inherent flaws in one-hot training (destructive gradients).
> - We provide **a novel formalization of the effect of early-stopping**. Early-stopping has been studied with great scrutiny in practice in the context of KD [(Dong et al., 2019; Cho & Hariharan, 2019; Ji & Zhu, 2020, Wang et al., 2022]. Importantly, our analysis yields an insight that opposes that of Mobahi et al., who say that early-stopping bias is very different from KD bias; we suggest that KD bias is an exaggerated version of early-stopping bias. This difference is important and stems from the fact that we look at the practically relevant GD setting.
> - We’ve also designed an extensive array of experiments to connect theory to practice and report results across (>6 different architectures and > 8 different image & language datasets) **which is not typical of the prior theory works we've built on**.
>
> Overall, while we do not progress all the way towards proving theories about deep nets, we hope you’ll agree that these three contributions, together with our new experiments involving deep nets constitute important progress towards practically-relevant theory. We sincerely hope you’ll reconsider your evaluation of our theoretical contributions in light of this.

---

> ### Author Response · Authors · 2022-12-07
> **Eager to hear your updated response**
>
> Dear Reviewer,
>
> We sincerely appreciate the time and effort you have taken to provide a detailed review. We've responded to each of your concerns individually in our response and extended our paper significantly with new results. The other three reviewers have acknowledged that these results address their concerns, including the concern regarding the linear nature of the theory, which you share. We are eager to hear your updated opinion of the paper  after reading our response! Thanks again for your time.
>
> Regards
>
> Authors

---

> > ### Comment · Reviewer_xDEX · 2022-12-10
> > **Thank you for the response**
> >
> >
> > Thank you for replying to most of my questions. I have read the rebuttal as well as other reviewer's comments and their responses.
> >
> > Below, I present some thoughts that lead me to believe the paper does not rise up to the level of acceptance and I would encourage the authors to incorporate some of the feedback into refining the next iteration of this work.
> >
> >
> > -- I believe the current structure of the paper, writing and lack of coherent/clear motivating story overshadows many important observations / key-points being studied in this work.
> >
> > -- Below, for completeness I have defined the One-hot and KD losses to make some observations and point out some flaws in the reasoning.
> >
> > One-Hot labels asks the student to minimize $-y \log s_y(x)$ where $y$ is the active label in the one-hot encoding and $s_y(x)$ denotes the student prediction probability for the active label
> >
> > While KD incorporates the following loss term $- \sum_y t_y(x) \log s_y(x)$ -- note that this incorporates all the classes $y$ with weight $t_y(x)$ -- in some sense denoting the softened teacher probability for the class $y$
> >
> > -- **Empirical Evaluations**:
> >  - Student tends to underfit on hard points for teacher -- why is this something surprising to begin with? If the teacher finds these data points hard, clearly the teacher probability distribution will have more than one active labels in the $t_y(x)$ vector. Since the transform being presented is $\log \frac{u}{1-u}$, there will be quite a few other classes that have non-trivial weight and as a result, the loss function asks the student to distinguish between these classes with not as high a weight on the teacher predicted class as the teacher had on the data points that are easy. This would result in student learning the predicted class with low score as seen in the figures.
> >
> >  - Effect of switching from one-hot to KD --
> >     - Point that destructive nature of one-hot training is simply that if you train long enough with one-hot labels after initial KD phase, it is bound to loose the inter-class relationship ( $t_y(x)$ ) soon enough that it learnt through KD, and will start to rely on the one-hot labels.
> >
> > 	- Point that most of the gains of full KD training can be recovered by first training with one-hot for 1/4 th epochs and then switching to KD.
> > 	Your learning rate starts at 1.0 and decays to 0.1 at 200 epoch. Total there are 450 epochs, so if you recover all the gains after switching after 1/4 epochs,i.e., 113 epochs, you are still starting with a high enough learning rate of 1.0 and learning with KD. 	If you switch at 1/2th epochs, i.e., after 250 epochs, the learning rate has not gone down to zero, but it has reduced to 0.1, so essentially it has juice in it to learn with the KD loss after the switch. It does come with some hit in performance that is visible from the figures.
> >
> >
> > -- **Theoretical Contributions** I appreciate the updated MNIST experiment and clarifications w.r.t. applicability of the linear setup. I have raised my score to reflect the same. I still feel that the setups in both the theoretical sections are too simple to explain the nuances of the KD process.
> >
> >
> > In the light of the above analysis, at best my score increases to 5, but it does not rise up to the level of acceptance. I strongly believe that the paper can be improved significantly such that some coherency can be extracted from the various empirical analysis and the theory can be perfected to more realistic setup.

---

> > > ### Author Response · Authors · 2022-12-10
> > > **Thanks for increasing your score!**
> > >
> > > Thank you for the acknowledgment that our experiments help address your concern regarding the applicability of the linear setup.
> > > Furthermore, thanks for patiently detailing your follow-up questions which we will try to address below.
> > >
> > > > Student tends to underfit on hard points for teacher -- why is this something surprising to begin with?
> > >
> > > **We wish to strongly emphasize our disagreement that this is unsurprising. Without the benefit of hindsight, it was not a priori clear that this is what one would observe from these scatter plots!** Since the KD loss minimizes $-\sum_y t_y(x) \log s_y(x)$, this minimum would be achieved when $s_y(x) = t_y(x)$. In such a case, these scatter plots should lie on the $X=Y$ line. There may be ``precision'' issues (e.g., due to learning rate being large and hence being unable to capture the probabilities exactly), but one would expect the resulting deviation to be uncharacteristic i.e., the deviations are likely to be on both sides of the $X=Y$ line and for all values of $X$.
> > >
> > > However, what we observe is generally a deviation _below the $X=Y$ line specifically for small values of $X$_. In hindsight, we interpret this as ``underfitting of hard points''. Given the notion of easy vs hard points, and given these scatter plots, this phenomenon may seem predictable, but perhaps, this is **hindsight bias**; without the notion of easy/hard points or these scatter plots, it was not obvious that this behavior would emerge. Furthermore, the mechanism underlying this is not obvious either, which we formalize via the eigenspace regularization effect.
> > >
> > > Besides, **we are afraid we do not follow the relevance of the transform $\log \frac{u}{1-u}$ in this discussion.** This is simply a monotonic transformation on $u$, so any argument based off of the transformed value, must apply to the original value as well. Please let us know if we're missing something here!
> > >
> > > > Switching from one-hot to KD...  you are still starting with a high enough learning rate of 1.0 and learning with KD
> > >
> > > It is worth noting that however late we start KD, we do find **an increase in accuracy over training time**. Please see Fig 1b  or Fig 16. Note that in these depicted plots, **the switch happens when/around the time when the learning rate has been decayed to the smallest value.** This observation is understandably not evident from the accuracy bar plots in Fig 14. Furthermore, we do admit that starting KD this late does not always regain all of the possible gains (although it does for CIFAR100 Res56 self-distillation!). Nevertheless, what is interesting is that there is a bump in accuracy.
> > >
> > >
> > > > [Switching from KD to one-hot]... it is bound to loose the inter-class relationship
> > >
> > > We must note that this again is not obvious. In the notations of Sec 4.2, it is possible that KD has learned a certain set of probabilities given by softmax$(\{\alpha \mathbf{f}_k({\mathbf{x}})\}\_{k=1}^K)$, and subsequent one-hot training simply scales up $\alpha\to\infty$ in order to fit the one-hot probabilities. **In such a hypothetical case, no inter-class relationship would be lost!** The idea that one-hot can _destroy_ gained information, is not empirically or theoretically obvious and not identified in prior literature to the best of our knowledge.
> > >
> > > > I still feel that the setups in both the theoretical sections are too simple to explain the nuances of the KD process.
> > >
> > > You're right that there may be more nuanced effects of KD that our linear setups do not capture; indeed, we admit such possibilities in the conclusion already! However, we wish to re-emphasize that it is fairer to evaluate these results **in terms of what novel insights they add on top of existing results** such as Menon et al., '21 and Mobahi et al., '20 **which do not apply to GD**. To rephrase, there are effects of KD we apparently did not understand in the GD + linear setting; formalizing these is a necessary first step before we attack the non-linear + GD setting.
> > >
> > > ------
> > >
> > >
> > > Again, we thank you for patiently engaging with us and for increasing your score. Your feedback has been valuable; we will continue to incorporate it for future iterations.

---

### Author Response · Authors · 2022-11-17
**Summary of response: Added experiments verifying theory in more general settings & clarification of takeaways**

We would like to thank the reviewers again for their detailed comments. Since we’ve taken the efforts to add many clarifying experiments, we’re summarizing them below.
## New experiments:
- We’ve **added a neural network + MNIST (Sec C.4) example where we show both eigenspace regularization and underfitting**, thus establishing that our insights from our linear theoretical model are still relevant to the deep learning setting.
- We’ve **added a quantitative experiment (Sec C.5) paralleling Thm 4.2 showing how distillation denoises** even in the absence of label noise. We also find underfitting in this setting, thus connecting our eigenspace view and gradient space view.
- We’ve added experiments **varying batch size and length of training** showing that the underfitting occurs even in these settings (Sec C.3)
- We show in Fig 16 that **persistent one-hot training** hurts the accuracy while distillation does not, thus clarifying that our theory about the destructive nature of one-hot gradients does hold.
- We added student-teacher scatter plots for **various other metrics such as KL-divergence, entropy** etc., (Fig 18).

## Outline of our work
As requested by one of our reviewers, here's a summary clarifying how our different findings coherently relate to each other.

The crux of our work is: While distillation is advocated as a process where the student mimics the teacher, we carefully identify what deviations exist between teacher and student training, how they arise, and when/why these deviations can be _good_:

- **what deviations exist?** Sec 3.1 identifies a novel type of student-teacher deviation: the underfitting of hard points.
- **which deviations are relevant?** Sec 3.2 notes that there are other deviations prior work have noted, but demonstrates that these other deviations are *not* critical to the success of distillation.
- **how do these deviations arise?** Sec 4.1 and 4.2 both provide a formalization of how optimization bias of GD causes deviations.
- **why are these deviations good?** Both the regularization and denoising effects in Sec 4.1 and 4.2 provide complementary insights into this question. Sec 4.3 hopes to provide a coherent picture of why these two insights (eigenspace regularization and gradient denoising) relate to each other.

As takeaways:
- **We argue that student-teacher deviations in probabilities can be good**, which has important implications for practitioners trying to bridge the teacher-student gap, an important area of distillation research.
- Understanding of distillation is still lacking and **our theory bridges significant gaps with practice by studying how distillation behaves under gradient descent, and in the absence of explicit label noise**. This requires non-trivial and novel arguments. In particular, it induces a regularization effect and a denoising effect.
- We also provide a **novel formalization of the importance of early-stopping in distillation**, which has been given significant attention in empirical work on distillation [(Dong et al., 2019; Cho & Hariharan, 2019; Ji & Zhu, 2020, Wang et al., 2022].

## Follow-up/practical suggestions
As requested by another one of the reviewers, we also wish to point out how our contributions can easily inspire both new theory and algorithmic work:
- One could explore ways to **modify distillation loss with importance weights** etc., to encourage the underfitting from Sec 3.1 and to amplify the eigenspace regularization.
-  Inspired by the loss-switching experiments in Sec 3.2, one could explore **different schedules of one-hot vs KD loss**.
- Using the framework of Sec 4.1, one can immediately **study variants of distillation** such as online/progressive distillation, and find ways to **further exaggerate the eigenspace bias**.
- Sec 4.2 suggests that there may be ways to improve one-hot training by **preventing the effect of destructive gradients**.

---

### Decision · Program_Chairs · 2023-01-20

**Decision:**

Reject

**Justification For Why Not Higher Score:**

Above summary should be clear.

**Justification For Why Not Lower Score:**

N/A

**Metareview: Summary, Strengths And Weaknesses:**

This paper studies the nature of deviations between teacher and student networks for knowledge distillation. The paper provides two different perspectives from the point of view of the eigenspace and from the point of view of the gradient space.

The opinions of the reviewers pre-rebuttal were very diverse. After the rebuttal, some reviewers increase their scores. However, the paper stayed a borderline case and was discussed with the meta AC.

Some important problems remain are:
1) The narrow setup of the theoretical results (Reviewers xDEX and M824). This is acknowledged by the authors (e.g. "Agreed that our theoretical analyses are in the linear setting")
2) Gap between theory and practice (Reviewer xDEX). This was addressed to some extent during the discussion period.
3) No clear takeaway from the paper, e.g. not clear what the value for practitioners would be (Reviewer EHbp)

- More minor: The theoretical results are not clearly compared to prior work. For instance, Phuong and Lampert derived similar results that involved a cosine similarity as well.

Overall, I think the first three points are important flaws that need to be fixed and I'm therefore not able to recommend acceptance.


**Summary Of Ac-Reviewer Meeting:**

I discussed this case with the meta AC and provided a summary above.

Unfortunately, I tried to set up a meeting with the reviewers but only two of them were responsive and they did not agree on a time slot suitable for both of them. I, therefore, discussed this case with the meta AC, and took the decision based on my own reading of the paper and reviews. I very much feel this paper is not ready for acceptance based on my own reading of the paper, and this is echoed by some of the reviewers as well. For instance, the reviewers who vouch for acceptance have also indicated that they will be okay with the rejection of the paper (no strong champions).